# Active Policy Optimization for Individualized Dosing via Gradient Variance Minimization

Yi Wan [1]  Xin Wang [2]  Huanhuan Chen [2]

## Abstract

In domains such as healthcare and marketing, learning optimal individualized dosing policies to maximize utility is crucial, yet high experimental costs impose strict budget constraints, necessitating efficient active policy learning. Existing active learning methods in causal inference primarily focus on binary treatments and effect estimation, leaving continuous dosing and policy optimization underexplored. To address this gap, we propose an active learning framework tailored for optimal policy learning. Exploiting the inherent structure of dose-response curves, we theoretically show that the policy optimization regret is bounded by the expected posterior gradient variance at the estimated optimal doses. Motivated by this result, we introduce Gradient Variance Active Learning for Individualized Dosing (GVALID), a batch acquisition strategy that greedily selects samples to minimize target gradient variance for efficient policy learning. Experiments demonstrate that GVALID achieves superior performance under strict budget constraints[1].

## 1. Introduction

Learning optimal personalized dosing policies is a central problem in precision medicine and individualized intervention (Murphy, 2003; Tucker, 2016). However, real-world interventions such as clinical trials are often costly and subject to stringent budget constraints. Traditional randomized controlled trials (RCTs) employ static, non-adaptive designs that struggle to efficiently explore vast continuous policy spaces within limited budgets. When data are inherently scarce, the research focus shifts from pursuing fitting accuracy to rapidly learning high-value dosing policies under budget constraints, thereby enabling the selection of utility-maximizing doses for future individuals.

To improve sample efficiency under budget constraints, a natural experimental paradigm is to partition the trial into multiple cohorts, where each cohort simultaneously executes a batch of interventions, observes feedback, and subsequently updates the model and allocation rule for the next cohort. This setting is closely related to pool-based batch active learning (Cohn et al., 1996) and batched contextual bandits (Gao et al., 2019), and has spurred numerous efficient exploration methods (Kalkanli & Ozgur, 2021; Li & Scarlett, 2022; Feng et al., 2022; Jiang & Ma, 2025). Concurrently, the integration of active learning with causal inference has deepened, spanning from query-based settings on observational data to experimental settings with controllable interventions (Jesson et al., 2021; Toth et al., 2022; Cha & Lee, 2025).

Despite significant progress, existing active causal learning research predominantly centers on effect estimation, focusing mainly on discrete (e.g., binary) treatments, with sampling criteria aimed at reducing estimation error or posterior variance of conditional effects (Puha et al., 2020; Cha & Lee, 2025). Even when extended to richer intervention spaces, common uncertainty-driven strategies still tend to improve the global predictive accuracy of dose-response functions (Razzak et al., 2024). However, policy optimization is not equivalent to global estimation: when the downstream objective is to find the optimal dose for each individual and control decision regret, allocating budget to clearly suboptimal dose regions for completing the fit often results in waste. This decision-focused objective connects our setting to best-arm identification (BAI) (Bubeck et al., 2011; Kato et al., 2024), but existing contextual BAI formulations mainly consider discrete treatment arms, leaving individualized continuous-dose policy learning less explored. Motivated by this (as illustrated in Figure 1), we aim to design a batch sampling mechanism for continuous dosing policy optimization under strict budget and one-shot constraints, directly aligning the sampling criterion with policy value

---

[1]School of Artificial Intelligence and Data Science, University of Science and Technology of China, Hefei, China [2]School of Computer Science and Technology, University of Science and Technology of China, Hefei, China. Correspondence to: Xin Wang <wxin0126@ustc.edu.cn>, Huanhuan Chen <hchen@ustc.edu.cn>.

*Proceedings of the 43rd International Conference on Machine Learning*, Seoul, South Korea. PMLR 306, 2026. Copyright 2026 by the author(s).

[1]The code is available at https://github.com/ry-wan/GVALID.

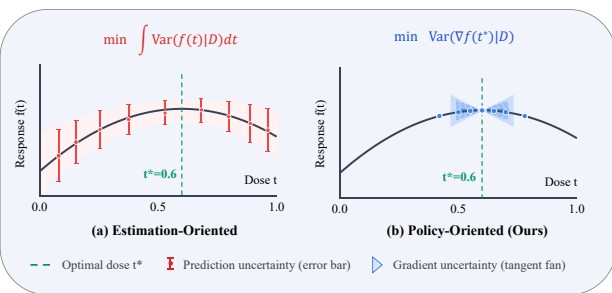

*Figure 1.* Intuitive comparison of sampling criteria. (a) Estimation-Oriented: Minimizes global predictive uncertainty (red error bars), leading to sample redundancy in sub-optimal regions. (b) Policy-Oriented (GVALID): Focuses on minimizing the gradient posterior variance at the estimated dose $t^*$ (blue tangent fans), thereby achieving superior sample efficiency by directly reducing uncertainty regarding stationarity.

maximization.

In real-world dose exploration, dose-response curves often exhibit certain regularities. For instance, in molecularly targeted drug and biologic trials, efficacy or biomarker responses typically vary smoothly with dose, and the trial objective is often to identify the optimal biological dose (OBD) in a unimodal sense (Zang et al., 2014; Taleb & West, 2023). Corresponding adaptive designs typically assume certain smoothness (Ren et al., 2020) in the dose dimension and rely on stable curvature near the optimum to ensure identifiability and learning stability of the OBD. Under this structural assumption, we prove that the Bayes regret upper bound of the posterior mean policy can be controlled by the gradient posterior variance at the policy-recommended dose. This result indicates that verifying whether a candidate dose is close to optimal hinges on reducing uncertainty about whether the gradient is zero (stationarity) in its vicinity, rather than indiscriminately reducing predictive uncertainty across the entire space.

Inspired by this insight, we propose GVALID (Gradient Variance Active Learning for Individualized Dosing), a batch sampling strategy. This method employs a GP surrogate model and constructs a gradient uncertainty objective around doses recommended by the current policy on a representative population. At each round, it selects a batch of interventions that maximally reduces the gradient variance at the target locations in expectation, thereby concentrating experimental resources on regions that most improve policy value. Experimental results demonstrate that, compared to estimation-oriented sampling baselines, this policy-oriented criterion achieves faster policy improvement under tight budgets. Our contributions are threefold:

• We propose a batch active learning framework for individualized dosing that directly aligns the sampling criterion with policy optimization rather than global effect estimation.

• We establish a theoretical bound linking policy regret to

gradient posterior variance and derive a practical objective to efficiently identify optimal doses.

• We demonstrate consistent advantages of our method across multiple budget-limited settings.

## 2. Related Work

This section contextualizes our work within three primary domains: individualized treatment rules, active learning, batched contextual bandits, and best-arm identification. We highlight key distinctions from existing methodologies below, deferring a more exhaustive discussion to Appendix A.

**Individualized Treatment Rules.** Our work contributes to the literature on Individualized Treatment Rules (ITRs). While a substantial body of research (Kallus & Zhou, 2018; Schweisthal et al., 2023; Xu et al., 2024; Sakaguchi, 2025) focuses on learning optimal policies from offline observational data, these approaches are inherently restricted to static logs and, by construction, remain agnostic to the constraints of active data acquisition. In contrast, we address an online, interventional setting. Rather than passive estimation, our framework actively governs the data generation process to directly optimize policy performance under a fixed experimental budget.

**Active Learning in Causal Inference.** Active learning (Cohn et al., 1996) improves data efficiency via adaptive sampling. Recently, this paradigm has been integrated into causal inference, spanning two primary directions based on the controllability of treatment assignment. The first targets fixed observational data, limiting the learner to querying labels for pre-assigned histories (Jesson et al., 2021; Wen et al., 2025). The second, aligned with our setting, enables active experimentation, where the learner controls interventions to generate new data (Puha et al., 2020; Cha & Lee, 2025; Toth et al., 2022; Razzak et al., 2024). Crucially, however, both directions prioritize minimizing *treatment effect estimation error* (e.g., via variance reduction). In contrast, we address the distinct task of *optimal policy learning*. Our selection mechanism is driven directly by policy value maximization, thereby avoiding the cumulative errors inherent in intermediate estimation steps.

**Batched Contextual Bandits.** Our work is closely related to the Batched Contextual Bandit (BCB) framework (Gao et al., 2019), which extends the standard bandit setting by enforcing batch constraints where feedback is observed only after a batch concludes. Recent advances in BCB have departed from parametric constraints toward nonparametric settings to capture complex nonlinear rewards. Notably, Feng et al. (2022) and Jiang & Ma (2025) established theoretical guarantees under smoothness assumptions, such as

Lipschitz continuity or Gaussian Process priors. Paralleling this trajectory, we adopt similar nonparametric priors for dose-response modeling. However, despite these shared functional assumptions, our work differs fundamentally in *learner agency*. While existing BCB methods operate under a passive regime with stochastic context arrivals, we enable *active context selection*, allowing for the strategic querying of informative individuals to enhance learning efficiency.

**Best-Arm Identification.** Our objective is also related to best-arm identification, where performance is evaluated by the quality of the final policy rather than cumulative reward (Bubeck et al., 2011; Kaufmann et al., 2016). Recent works further extend this perspective to linear, transductive, and contextual policy-learning settings (Fiez et al., 2019; Jedra & Proutiere, 2020; Kato et al., 2024). While these works share our decision-focused objective, they typically consider finite or linearly parameterized arms. In contrast, we study individualized continuous dosing under one-shot batched experimentation, and exploit smooth dose-response geometry through a tractable gradient-variance surrogate.

## 3. Problem Formulation

For each individual $i$, let $\mathbf{x}_i \in \mathcal{X}$ denote its covariate vector, where $\mathcal{X} \subseteq \mathbb{R}^d$ denotes the feasible covariate space. Let the continuous dose space be $\mathcal{T} = [0, 1]$. For each individual $\mathbf{x} \in \mathcal{X}$ and dose $t \in \mathcal{T}$, denote the potential outcome (Rubin, 1974) by $Y_{\mathbf{x}}^t$. The individualized dose-response function (conditional potential outcome function) is defined as: $f(x, t) = \mathbb{E}[Y_{\mathbf{x}}^t \mid X = \mathbf{x}]$, so that under an intervention assigning dose $t$ to $\mathbf{x}$ we observe

$$Y_{\mathbf{x}}^t = f(\mathbf{x}, t) + \varepsilon, \qquad \varepsilon \sim \mathcal{N}(0, \sigma_\epsilon^2), \qquad (1)$$

with finite noise variance $\sigma_\epsilon^2$.

**Identification assumptions.** We adopt the standard causal assumptions commonly used to identify dose-response functions, which can be referred in Appendix B.

**Individualized treatment rule.** Given the identified dose-response function, we learn individualized decision rules. An individualized treatment rule, or dosing policy, is defined as a measurable function $\pi : \mathcal{X} \to \mathcal{T}$. The value of any given policy $\pi$ is defined as its expected outcome $V(\pi) := \mathbb{E}_X[f(X, \pi(X))]$. An optimal policy $\pi^*$ in a given policy class $\Pi$ is defined as the one that maximizes $V(\pi)$, i.e.

$$\pi^* \in \arg\max_{\pi \in \Pi} V(\pi). \qquad (2)$$

**Batched active experimentation with one-shot interventions.** We study an adaptive experimental design for learning an ITR. Our setting parallels *pool-based batch active learning*, where the learner maintains (i) a pool of available, unlabeled individuals $\mathcal{X}_{\text{pool}}$ and (ii) an ever-growing

set $\mathcal{X}_{\text{train}}$ obtained by querying an oracle for outcomes of selected interventions. The experiment runs for at most $T$ rounds. At round $\tau$, we select a batch of intervention queries $\mathcal{S}^\tau = \{(\mathbf{x}_j, t_j)\}_{j=1}^B$ from the set of not-yet-intervened individuals $\mathcal{X}_{\text{avail}} := \mathcal{X}_{\text{pool}} \setminus \mathcal{X}_{\text{train}}$, execute these interventions, and observe outcomes $\{y_j\}_{j=1}^B$. Let $\mathcal{D}^\tau$ denote the cumulative dataset collected up to (and including) round $\tau$, i.e., $\mathcal{D}^\tau := \{(\mathbf{x}_j, t_j, y_j)\}_{j \in \mathcal{I}^\tau}$, where $\mathcal{I}^\tau$ indexes all queried interventions up to round $\tau$. We then update the dataset by appending the newly acquired batch, $\mathcal{D}^{\tau+1} := \mathcal{D}^\tau \cup \{(\mathbf{x}_j, t_j, y_j)\}_{j=1}^B$.

The batch selection is governed by an adaptive acquisition rule $\rho^\tau$ that depends only on past data $\mathcal{D}^\tau$ and the current pool. At each round, the augmented dataset is used to update the policy and to construct the next acquisition rule. Crucially, each individual can be queried at most once (one-shot constraint), i.e., once $\mathbf{x}$ is selected in some batch it is removed from the pool, reflecting practical regimes where repeated experimentation on the same unit is infeasible or undesirable (Robertson et al., 2023).

**Design objective.** Under a total budget of at most $TB$ interventions and the one-shot constraint, our design goal is to design the adaptive assignment strategy $\{\rho^\tau\}_{\tau=1}^T$ so as to output a policy $\hat{\pi}$ with maximal value $V(\hat{\pi})$, equivalently minimizing the regret

$$R(\hat{\pi}) = V(\pi^*) - V(\hat{\pi}). \qquad (3)$$

## 4. Variance-Based Objective under Gaussian Processes

To identify the most informative interventions, a common approach is to adopt a Bayesian surrogate and prioritize candidates with high posterior uncertainty. Gaussian Processes (GPs) instantiate this principle by placing a nonparametric Bayesian prior over the dose–response function and providing closed-form predictive variances. This section first presents the GP-based estimator and its induced posterior mean policy. It then relates regret bounds to GP posterior uncertainty, motivating our acquisition criterion.

### 4.1. Gaussian Process as surrogate model

We first assume that the dose-response function $f : \mathcal{X} \times \mathcal{T} \to \mathbb{R}$ satisfies the following Assumptions 4.1 and 4.2.

**Assumption 4.1** (Smoothness). Let $g(\mathbf{x}, t) \triangleq \partial_t f(\mathbf{x}, t)$. $\forall (\mathbf{x}, t) \in \mathcal{X} \times \mathcal{T}$: 1) $\exists\, 0 < \mu \leq L < \infty$, $-\partial_{tt} f(\mathbf{x}, t) \in [\mu, L]$. 2) $\exists\, G_m < \infty$, such that $|g(\mathbf{x}, t)| \leq G_m$.

**Assumption 4.2** (Gaussian Process Prior). Let $f$ be assigned a Gaussian Process prior $f \sim \mathcal{GP}(0, k)$, where the kernel $k((\mathbf{x}, t), (\mathbf{x}', t'))$ is twice continuously differentiable in dose space. Then the dose gradient $g(\mathbf{x}, t)$ exists in the mean-square sense and is itself a Gaussian Process.

In continuous dosing settings, studies of biologics or molecularly targeted agents often reveal a non-monotonic dose–response relationship with a unimodal optimal biological dose (OBD). This motivates introducing moderate smoothness and shape constraints along the dose dimension to ensure OBD identifiability (Zang et al., 2014; Ren et al., 2020). Likewise, nonparametric Bayesian methods such as Gaussian Processes, commonly applied in continuous dose–response modeling and sequential decision-making, rely on dose-level smoothness and differentiability for stable learning and uncertainty quantification (Hamza et al., 2021; Li & Laber, 2025). Accordingly, we impose a standardized smoothness constraint on the dose dimension in Assumption 4.1 to rule out unrealistic spiky responses, and specify in Assumption 4.2 a Gaussian Process prior as a Bayesian surrogate for learning $f$ from noisy, adaptively collected interventional data.

**GP surrogate model and the induced policy.** Given any dataset $\mathcal{D} = \{(\mathbf{x}_i, t_i, y_i)\}_{i=1}^n$ generated by the mechanism in Equation (1), Assumption 4.2 implies that for any query $(\mathbf{x}, t)$ the posterior distribution $p(f(\mathbf{x}, t) \mid \mathcal{D})$ is Gaussian. We denote its posterior mean and covariance by

$$\mu_{\mathcal{D}}(\mathbf{x}, t) := \mathbb{E}[f(\mathbf{x}, t) \mid \mathcal{D}], \tag{4}$$

$$k_{\mathcal{D}}((\mathbf{x}, t), (\mathbf{x}', t')) := \mathrm{Cov}(f(\mathbf{x}, t), f(\mathbf{x}', t') \mid \mathcal{D}). \tag{5}$$

We extract a deterministic individualized dosing rule by maximizing the posterior mean over the dose space:

$$\hat{t}_{\mathcal{D}}^*(\mathbf{x}) \in \hat{\pi}_{\mathcal{D}}(\mathbf{x}) := \arg\max_{t \in \mathcal{T}} \mu_{\mathcal{D}}(\mathbf{x}, t). \tag{6}$$

We assume $\hat{t}_{\mathcal{D}}^*(\mathbf{x})$ lies in the interior of $\mathcal{T}$, thereby satisfying the first-order optimality condition (see Appendix B for details). Building on the regret defined in Equation (3), the posterior-mean policy induces the posterior-conditional regret

$$R(\mathcal{D}) := \mathbb{E}\left[V(\pi^*) - V(\hat{\pi}_{\mathcal{D}}) \mid \mathcal{D}\right], \tag{7}$$

which we refer to as the *Bayes regret*, quantifying the remaining performance gap under the posterior uncertainty at deployment time.

## 4.2. Surrogate objective for policy optimization

This subsection develops a tractable surrogate objective for policy optimization by translating the regret–dose relationship into a quantity that can be evaluated and optimized using bandit feedback. First, we present a lemma that provides a direct, intuitive characterization.

**Lemma 4.1** (Local Regret-Dose Relationship). *Under Assumption 4.1, $\forall \mathbf{x} \in \mathcal{X}$, let $t^*(\mathbf{x}) := \arg\max_{t \in \mathcal{T}} f(\mathbf{x}, t)$ be the optimal dose, and define the pointwise regret at dose $t$ as $R(\mathbf{x}; t) := f(\mathbf{x}, t^*(\mathbf{x})) - f(\mathbf{x}, t)$. Then, for all $t \in \mathcal{T}$:*

$$R(\mathbf{x}; t) \leq \frac{L}{2}|t - t^*(\mathbf{x})|^2, \tag{8}$$

*where $L$ is defined in Assumption 4.1.*

Lemma 4.1 establishes that for any fixed individual $\mathbf{x}$, the policy's pointwise regret at $\mathbf{x}$ scales quadratically with the deviation between the administered dose and the optimal dose. Consequently, a small dose estimation error implies a small pointwise regret.

However, while Lemma 4.1 is naturally expressed in terms of the dose gap $|t - t^*(\mathbf{x})|$, the optimizer $t^*(\mathbf{x})$ is defined only implicitly via $t^*(\mathbf{x}) \in \arg\max_t f(\mathbf{x}, t)$ and is neither directly observable nor directly controllable under bandit feedback. This consideration motivates a gradient-based perspective: under Assumption 4.1, the local gradient magnitude $|\partial_t f(\mathbf{x}, t)|$ provides an informative surrogate for $|t - t^*(\mathbf{x})|$ in a neighborhood of $t^*(\mathbf{x})$, leading to the gradient characterization formalized in Theorem 4.1.

**Theorem 4.1** (Uncertainty Regret Bound). *Under Assumptions 4.1 and 4.2 and the posterior mean policy Equation (6),*

$$R(\mathcal{D}) \leq C_{\mathrm{g}} \cdot \mathbb{E}_X\left[\mathrm{Var}\left(g(X, \hat{t}_{\mathcal{D}}^*(X)) \mid \mathcal{D}\right)\right], \tag{9}$$

*where $C_{\mathrm{g}} := \frac{L}{2\mu^2}$, $\mu$, $L$ from Assumption 4.1 1).*

Theorem 4.1 shows that the population-level regret is controlled by the integrated posterior variance of the gradients of the policy $\hat{t}_{\mathcal{D}}^*(\mathbf{x})$. This directly motivates the design objective of GVALID, which aims to reduce this integrated variance as quickly as possible.

In OBD studies, dose-response curves may also exhibit plateau-shaped behavior, where the expected response saturates beyond a certain dose and further escalation may increase toxicity risk (Guo & Yuan, 2023). Thus, the optimal dose can naturally form a flat interval rather than a singleton. The following corollary shows that our gradient-variance argument extends to this setting.

**Corollary 4.1** (Flat Optimal Regions). *For each $\mathbf{x}$, let $\mathcal{T}^\star(\mathbf{x}) := \arg\max_{t \in \mathcal{T}} f(\mathbf{x}, t)$ be an interval-valued optimal dose set, i.e., $\mathcal{T}^\star(\mathbf{x}) = [a(\mathbf{x}), b(\mathbf{x})]$ with $a(\mathbf{x}) \leq b(\mathbf{x})$. Assume that $f(\mathbf{x}, t)$ satisfies the set-valued regret–gradient condition: for all $t^\star \in \mathcal{T}^\star(\mathbf{x})$, and for all $t \in \mathcal{T}$,*

$$f(\mathbf{x}, t^\star) - f(\mathbf{x}, t) \leq \frac{L}{2}\mathrm{dist}^2(t, \mathcal{T}^\star(\mathbf{x})) \leq \frac{L}{2\mu^2}|g(\mathbf{x}, t)|^2,$$

*where $\mathrm{dist}(t, \mathcal{T}^\star(\mathbf{x})) := \inf_{s \in \mathcal{T}^\star(\mathbf{x})}|t - s|$. Then, under Assumption 4.2, the regret bound in Theorem 4.1 remains valid after replacing the unique optimum $t^\star(\mathbf{x})$ with the optimal set $\mathcal{T}^\star(\mathbf{x})$.*

Therefore, the proposed acquisition objective is not limited to peaked unimodal responses. For plateau-shaped OBD responses, minimizing gradient uncertainty can still guide the policy toward the flat optimal region.

**Algorithm 1** GVALID (Batch Acquisition)
___
**Input:** outer round $\tau$, GP posterior $(\mu^\tau, k^\tau)$, pool $\mathcal{X}_{\text{pool}}$ with available set $\mathcal{X}_{\text{avail}}$, batch size $B$, grid $\mathcal{T}_c$, target size $m$, candidate size $n_c$, finite-difference step $\delta$.
**Output:** Batch $\mathcal{S}^\tau = \{(\mathbf{x}_b, t_b)\}_{b=1}^B \subseteq (\mathcal{X}_{\text{avail}} \times \mathcal{T})$.
Sample targets $\{\mathbf{x}_j\}_{j=1}^m$ from $\mathcal{X}_{\text{pool}}$.
**for** $j = 1$ **to** $m$ **do**
    $t_j \leftarrow \arg\max_{t \in \mathcal{T}} \mu^\tau(\mathbf{x}_j, t)$.
    Set $(t_j^-, t_j^+) \leftarrow \text{clip}(t_j \pm \delta, \mathcal{T})$.
    Construct the finite-difference $\tilde{g}_j$ via Equation (11).
**end for**
Subsample candidates $\mathcal{X}_c = \{\mathbf{x}_i\}_{i=1}^{n_c}$ from $\mathcal{X}_{\text{avail}}$.
Form candidate set $\mathcal{Q}^\tau$ according to Equation (14).
Initialize selected set $\mathcal{S}^\tau \leftarrow \emptyset$.
**repeat**
    $\forall q \in \mathcal{Q}^\tau$, compute the score $\Delta^{\mathcal{S}}_{\mathcal{D}_\tau}(q)$ via Equation (16).
    Select $q^\star \leftarrow \arg\max_{q \in \mathcal{Q}^\tau} \Delta^{\mathcal{S}}_{\mathcal{D}_\tau}(q)$.
    $\mathcal{S}^\tau \leftarrow \mathcal{S}^\tau \cup \{q^\star\}$.
    Remove $q^\star$ from $\mathcal{Q}^\tau$ (one-shot constraint).
    Update posterior covariance quantities conditioned on $q^\star$ via Equations (18) and (19).
**until** $|\mathcal{S}^\tau| = B$ **or** $\mathcal{Q}^\tau = \emptyset$
___

# 5. Method

Section 4 shows that the optimization regret admits an upper bound governed by the posterior uncertainty of the dose gradient. Accordingly, our active learning framework is designed to iteratively reduce this gradient uncertainty across rounds. To achieve the largest per-round decrease of this objective, we propose GVALID, a batch acquisition strategy that selects the intervention queries expected to maximally reduce the gradient-variance criterion. We next introduce GVALID and present the full active learning loop that alternates data acquisition, model updating, and policy updates.

## 5.1. Finite-difference gradient proxies

GVALID first randomly samples a representative context set to estimate the gradient variance under the current policy, then greedily selects a batch from a subsampled candidate pool to minimize this variance-based objective.

**Finite target approximation.** Our surrogate objective in Theorem 4.1 involves the population quantity $\mathbb{E}_X \big[ \text{Var}\big(g(X, t^*_{\mathcal{D}}(X)) \mid \mathcal{D}\big) \big]$, which is generally intractable to evaluate exactly. We therefore approximate it using a finite set of representative contexts sampled from the deployment pool. At round $\tau$, we draw $\mathcal{X}^\tau = \{\mathbf{x}_j\}_{j=1}^m$, $\mathbf{x}_j \sim \text{U}(\mathcal{X}_{\text{pool}})$, and form the target set under the current policy $t_j := \hat{\pi}^\tau(\mathbf{x}_j)$, $\mathcal{Z}^\tau := \{\mathbf{z}_j = (\mathbf{x}_j, t_j)\}_{j=1}^m$. The

resulting finite-dimensional uncertainty objective is

$$\Phi_{\mathcal{D}_\tau}(\mathcal{Z}^\tau) := \sum_{j=1}^m \text{Var}\big(g(\mathbf{x}_j, t_j) \mid \mathcal{D}^\tau\big). \quad (10)$$

It can be shown that minimizing Equation (10) tightens the resulting upper bound on the Bayes policy regret, up to a finite-sample approximation error. This theoretical guaranty is provided in Appendix C, Theorem C.1.

**Finite-difference construction.** Since the true gradient $g(\mathbf{x}_j, t_j)$ is not directly available, we approximate it via a symmetric finite-difference scheme (Kiefer & Wolfowitz, 1952), thereby controlling gradient uncertainty only on $\mathcal{Z}^\tau$. Fix a step size $\delta > 0$ and choose $\mathbf{z}_j^+ = (\mathbf{x}_j, t_j^+)$, $\mathbf{z}_j^- = (\mathbf{x}_j, t_j^-)$, $t_j^+ - t_j^- = 2\delta$, with $t_j^-, t_j^+ \in \mathcal{T}$ enforced in the implementation. Define the finite-difference gradient proxy

$$\tilde{g}_j := \frac{f(\mathbf{z}_j^+) - f(\mathbf{z}_j^-)}{2\delta}, \quad \tilde{\mathbf{g}} := (\tilde{g}_1, \dots, \tilde{g}_m)^\top. \quad (11)$$

Collect the $2m$ function values at the finite-difference locations: $\mathbf{f}_{\text{fd}} := \big(f(\mathbf{z}_1^+), \dots, f(\mathbf{z}_m^+), f(\mathbf{z}_1^-), \dots, f(\mathbf{z}_m^-)\big)^\top \in \mathbb{R}^{2m}$. There exists a fixed matrix $\mathbf{A} \in \mathbb{R}^{m \times 2m}$ (determined by $\delta$ and the pairing of $(\mathbf{z}_j^+, \mathbf{z}_j^-)$) such that $\tilde{\mathbf{g}} = \mathbf{A}\mathbf{f}_{\text{fd}}$. Under the GP model, $\mathbf{f}_{\text{fd}} \mid \mathcal{D}^\tau \sim \mathcal{N}(\boldsymbol{\mu}^\tau, \Sigma^\tau)$, where we abbreviate $\boldsymbol{\mu}^\tau$ and $\Sigma^\tau$ for the GP posterior mean and covariance conditioned on $\mathcal{D}^\tau$. Therefore, by the closure of linear-Gaussian transformations,

$$\tilde{\mathbf{g}} \mid \mathcal{D}^\tau \sim \mathcal{N}(\mathbf{A}\boldsymbol{\mu}^\tau, \Sigma_{\tilde{g}}^\tau), \quad \Sigma_{\tilde{g}}^\tau := \mathbf{A}\Sigma^\tau \mathbf{A}^\top. \quad (12)$$

We thus use the trace objective

$$\tilde{\Phi}_{\mathcal{D}_\tau}(\mathcal{Z}^\tau) := \text{tr}\big(\Sigma_{\tilde{g}}^\tau\big) = \sum_{j=1}^m \text{Var}(\tilde{g}_j \mid \mathcal{D}^\tau), \quad (13)$$

as a computable surrogate for the aggregate posterior uncertainty of the gradients at the current policy, which will be minimized by our query selection rule in the next subsection.

## 5.2. Greedy batch selection with Schur complement updates.

At each round $\tau$, we construct a one-shot batch of interventions by (i) restricting attention to a finite candidate set $\mathcal{Q}^\tau$ and (ii) greedily maximizing the trace-reduction objective, with rank-one Schur-complement updates enabling efficient covariance refresh after each selection.

**Candidate query set.** Given the one-shot constraint, we only consider interventions on yet-unqueried individuals. We first randomly subsample $n_c$ candidate individuals $\mathcal{X}_c := \{\mathbf{x}_i\}_{i=1}^{n_c}$ from the available pool $\mathcal{X}_{\text{avail}}$. For each

candidate $\mathbf{x}_i$, we partition the dose space into a collection of cells $\{I_k\}_{t_k \in \mathcal{T}_c}$ centered at each grid point of a coarse grid $\mathcal{T}_c$. We then form the candidate query set by stochastically sampling dosages from these cells:

$$\mathcal{Q}^\tau := \{q = (\mathbf{x}, t) : \mathbf{x} \in \mathcal{X}_c, t \sim \mathrm{U}(I_k), t_k \in \mathcal{T}_c\}. \quad (14)$$

**Objective: trace reduction.** Given the feasible query set $\mathcal{Q}^\tau$, we formulate batch selection by quantifying the reduction in the trace-based uncertainty functional $\tilde{\Phi}(\cdot)$ (as defined in Equation (10)) under look-ahead conditioning, i.e., by treating a candidate set of query locations $\mathcal{S} \subseteq \mathcal{Q}^\tau$ as if their outcomes were observed and measuring the resulting decrease in posterior uncertainty. Specifically, for any $\mathcal{S} \subseteq \mathcal{Q}^\tau$ we define the look-ahead trace objective

$$\tilde{\Phi}_{\mathcal{D}^\tau}^{\mathcal{S}}(\mathcal{Z}^\tau) := \sum_{j=1}^m \mathrm{Var}(\tilde{g}_j \mid \mathcal{D}^\tau, \mathcal{S}), \quad (15)$$

and, when the context is clear, we suppress the dependence on $\mathcal{Z}^\tau$ and write $\tilde{\Phi}_{\mathcal{D}^\tau}^{\mathcal{S}}$. Then, the following Theorem 5.1 derives a closed-form expression for the one-step marginal reduction in the look-ahead trace objective induced by any additional candidate query $q \in \mathcal{Q}^\tau$.

**Theorem 5.1** (Single-Observation Schur Reduction). *Under Assumption 4.2 and the finite-difference construction in Equation (12), fix $\mathcal{Z}^\tau$ and $\tilde{\mathbf{g}}$. Given $\mathcal{D}^\tau$, for any $q \in \mathcal{Q}^\tau$, let $y_q = f(q) + \varepsilon$ with $\varepsilon \sim \mathcal{N}(0, \sigma_\epsilon^2)$. For any (possibly empty) set $\mathcal{S} \subseteq \mathcal{Q}^\tau$, define the look-ahead posterior quantities*

$$\mathbf{c}_{\mathcal{D}^\tau}^{\mathcal{S}}(q) := \mathrm{Cov}(\tilde{\mathbf{g}}, y_q \mid \mathcal{D}^\tau, \mathcal{S}), \ s_{\mathcal{D}^\tau}^{\mathcal{S}}(q) := \mathrm{Var}(y_q \mid \mathcal{D}^\tau, \mathcal{S}),$$

*then the one-step trace reduction from adding $q$ on top of $\mathcal{S}$ satisfies*

$$\Delta_{\mathcal{D}^\tau}^{\mathcal{S}}(q) := \tilde{\Phi}_{\mathcal{D}^\tau}^{\mathcal{S}} - \tilde{\Phi}_{\mathcal{D}^\tau}^{\mathcal{S} \cup \{q\}} = \frac{\left\| \mathbf{c}_{\mathcal{D}^\tau}^{\mathcal{S}}(q) \right\|_2^2}{s_{\mathcal{D}^\tau}^{\mathcal{S}(q)}}. \quad (16)$$

By Theorem 5.1, each candidate $q \in \mathcal{Q}^\tau$ admits a closed-form one-step trace-reduction score. Thus, a natural and direct strategy is to select a batch $\mathcal{S} \subset \mathcal{Q}^\tau$ of size $B$ that maximizes the cumulative reduction in the trace objective—equivalently, that minimizes the remaining posterior uncertainty:

$$\mathcal{S}^* \in \arg \max_{\mathcal{S} \subset \mathcal{Q}^\tau, |\mathcal{S}| = B} \left( \tilde{\Phi}_{\mathcal{D}^\tau}^{\emptyset} - \tilde{\Phi}_{\mathcal{D}^\tau}^{\mathcal{S}} \right). \quad (17)$$

**Greedy batch selection.** Since (17) is combinatorial, we adopt a greedy approximation (Nemhauser et al., 1978) that constructs $\mathcal{S}$ sequentially. Starting from $\mathcal{S} = \emptyset$, we repeatedly add the candidate with the largest marginal reduction $\Delta_{\mathcal{D}^\tau}^{\mathcal{S}}(q)$ until $|\mathcal{S}| = B$. Every time, after selecting $q^\star$, the look-ahead covariance and variance terms for any remaining

$q$ can be updated via rank-one Schur-complement updates (Gill et al., 1974):

$$\mathbf{c}_{\mathcal{D}^\tau}^{\mathcal{S} \cup \{q^\star\}}(q) = \mathbf{c}_{\mathcal{D}^\tau}^{\mathcal{S}}(q) - \frac{\mathbf{c}_{\mathcal{D}^\tau}^{\mathcal{S}}(q^\star) \mathrm{Cov}(y_{q^\star}, y_q \mid D^\tau, \mathcal{S})}{s_{\mathcal{D}^\tau}^{\mathcal{S}}(q^\star)}, \quad (18)$$

$$s_{\mathcal{D}^\tau}^{\mathcal{S} \cup \{q^\star\}}(q) = s_{\mathcal{D}^\tau}^{\mathcal{S}}(q) - \frac{\mathrm{Cov}(y_{q^\star}, y_q \mid \mathcal{D}^\tau, \mathcal{S})^2}{s_{\mathcal{D}^\tau}^{\mathcal{S}}(q^\star)}. \quad (19)$$

Thus, the scores $\Delta_{\mathcal{D}^\tau}^{\mathcal{S}}(\cdot)$ can be refreshed efficiently without recomputing the full posterior covariance from scratch. The selection process is depicted in Algorithm 1.

**Active learning loop.** Following each round, the dataset $\mathcal{D}_{\tau+1} = \mathcal{D}_\tau \cup \mathcal{S}^\tau$ is used to update the GP posterior and the resulting target policy $\hat{\pi}$. This cycle repeats iteratively; see Appendix E for the full algorithmic framework.

## 6. Experiments

In this section, we conduct experiments to validate the efficacy of our proposed framework. We evaluate our algorithm on three synthetic datasets, with dose-response functions ranging from simple to complex shapes, as well as on the semi-synthetic benchmark News. For a detailed description of the datasets, please refer to Appendix F. We partition the data into a training pool and a disjoint test set (7:3 ratio), enforcing strict physical isolation to deny the model any access to test information during the active learning loop. To ensure a rigorous comparison, all baselines share an identical test set per seed, with the total acquisition budget fixed at 20% of the training pool to evaluate sample efficiency in label-scarce regimes.

We compare our approach against three categories of baselines adapted for the individualized dosing task in batch settings: (i) active learning strategies (MaxVar (Seo et al., 2000), Soft Top-K (Kirsch et al., 2023)); (ii) batch contextual bandit algorithms (TS (Kalkanli & Ozgur, 2021), GP-UCB (Li & Scarlett, 2022), EI (Gupta et al., 2022), PLAS (Kato et al., 2024), and CATS (Majzoubi et al., 2020)); and (iii) methods specifically designed for causal inference (ABC3 (Cha & Lee, 2025) and PG (Razzak et al., 2024)), along with a random baseline (RAND). For a rigorous comparison, all baselines utilize a unified Gaussian Process framework and consistent global settings, such as batch size and initialization. We have tailored strategies originally restricted to context-only or treatment-only selection to accommodate our joint acquisition setting. Furthermore, extensive hyperparameter searches were conducted for each method to ensure peak performance. Implementation details and adaptation logic are provided in Appendix G. We evaluate our method using policy suboptimality $\mathcal{R}$, defined as the expected regret: $\mathcal{R} := \mathbb{E}\left[f(\mathbf{x}, t^*) - f(\mathbf{x}, \hat{t})\right]$, where $t^*$ and

*Table 1.* Policy suboptimality ($\mathcal{R}$) on four simulated datasets at representative query budgets ($N$). A complete evaluation over additional update steps is provided in Appendix H.1. (STS results are scaled by $10^{-2}$, and standard deviations are shown as subscripts.)

| METHOD | $N = 150$ | | | | $N = 250$ | | | | $N = 350$ | | | |
|---|---|---|---|---|---|---|---|---|---|---|---|---|
| | HNL | CSC | SWV | STS | HNL | CSC | SWV | STS | HNL | CSC | SWV | STS |
| TS | $0.56_{\pm.21}$ | $1.15_{\pm.52}$ | $0.79_{\pm.20}$ | $1.19_{\pm1.27}$ | $0.39_{\pm.12}$ | $0.87_{\pm.42}$ | $0.56_{\pm.16}$ | $0.89_{\pm.96}$ | $0.31_{\pm.11}$ | $0.72_{\pm.34}$ | $0.43_{\pm.11}$ | $0.64_{\pm.83}$ |
| GP-UCB | $0.78_{\pm.38}$ | $1.78_{\pm1.7}$ | $0.97_{\pm.33}$ | $0.96_{\pm1.00}$ | $0.58_{\pm.27}$ | $1.34_{\pm1.1}$ | $0.74_{\pm.21}$ | $0.77_{\pm1.09}$ | $0.46_{\pm.16}$ | $0.91_{\pm.43}$ | $0.57_{\pm.15}$ | $0.54_{\pm.55}$ |
| EI | $0.90_{\pm.35}$ | $2.45_{\pm2.9}$ | $1.13_{\pm.39}$ | $1.27_{\pm1.56}$ | $0.68_{\pm.23}$ | $2.38_{\pm3.4}$ | $0.76_{\pm.17}$ | $0.56_{\pm.90}$ | $0.55_{\pm.20}$ | $2.08_{\pm3.3}$ | $0.62_{\pm.13}$ | $0.61_{\pm.78}$ |
| CATS | $0.71_{\pm.27}$ | $1.26_{\pm.62}$ | $0.84_{\pm.29}$ | $1.21_{\pm1.03}$ | $0.43_{\pm.11}$ | $0.95_{\pm.65}$ | $0.56_{\pm.20}$ | $0.53_{\pm.41}$ | $0.31_{\pm.08}$ | $0.66_{\pm.28}$ | $0.38_{\pm.14}$ | $0.28_{\pm.38}$ |
| MAXVAR | $0.41_{\pm.17}$ | $0.87_{\pm.41}$ | $0.96_{\pm.42}$ | $0.59_{\pm.59}$ | $0.25_{\pm.09}$ | $0.65_{\pm.26}$ | $0.52_{\pm.23}$ | $0.42_{\pm.45}$ | $0.18_{\pm.06}$ | $0.50_{\pm.18}$ | $0.35_{\pm.11}$ | $0.25_{\pm.42}$ |
| ABC3 | $0.43_{\pm.15}$ | $0.86_{\pm.40}$ | $0.79_{\pm.28}$ | $1.13_{\pm1.37}$ | $0.24_{\pm.09}$ | $0.67_{\pm.25}$ | $0.43_{\pm.14}$ | $0.37_{\pm.54}$ | $0.17_{\pm.05}$ | $0.55_{\pm.18}$ | $0.36_{\pm.10}$ | $0.17_{\pm.40}$ |
| PG | $0.54_{\pm.18}$ | $0.91_{\pm.38}$ | $0.67_{\pm.16}$ | $1.44_{\pm1.14}$ | $0.40_{\pm.09}$ | $0.63_{\pm.22}$ | $0.50_{\pm.11}$ | $1.18_{\pm1.04}$ | $0.33_{\pm.07}$ | $0.51_{\pm.16}$ | $0.38_{\pm.11}$ | $1.06_{\pm1.00}$ |
| SOFTTOPK | $0.43_{\pm.17}$ | $1.57_{\pm2.6}$ | $0.84_{\pm.37}$ | $\mathbf{0.56}_{\pm.38}$ | $0.25_{\pm.09}$ | $1.22_{\pm2.6}$ | $0.51_{\pm.20}$ | $0.73_{\pm.70}$ | $0.18_{\pm.06}$ | $1.08_{\pm2.4}$ | $0.35_{\pm.10}$ | $0.66_{\pm.54}$ |
| PLAS | $0.49_{\pm.17}$ | $0.94_{\pm.37}$ | $\mathbf{0.60}_{\pm.07}$ | $0.98_{\pm.86}$ | $0.41_{\pm.12}$ | $0.69_{\pm.28}$ | $0.39_{\pm.15}$ | $0.91_{\pm.96}$ | $0.33_{\pm.09}$ | $0.53_{\pm.19}$ | $0.34_{\pm.11}$ | $0.93_{\pm.98}$ |
| RAND | $0.54_{\pm.16}$ | $1.00_{\pm.62}$ | $0.71_{\pm.23}$ | $0.88_{\pm.72}$ | $0.40_{\pm.11}$ | $0.71_{\pm.31}$ | $0.43_{\pm.09}$ | $0.88_{\pm.86}$ | $0.33_{\pm.10}$ | $0.56_{\pm.22}$ | $0.37_{\pm.08}$ | $0.78_{\pm.87}$ |
| GVALID | $\mathbf{0.40}_{\pm.24}$ | $\mathbf{0.84}_{\pm.53}$ | $0.65_{\pm.19}$ | $0.76_{\pm.92}$ | $\mathbf{0.18}_{\pm.09}$ | $\mathbf{0.50}_{\pm.32}$ | $\mathbf{0.37}_{\pm.12}$ | $\mathbf{0.36}_{\pm.47}$ | $\mathbf{0.12}_{\pm.04}$ | $\mathbf{0.40}_{\pm.26}$ | $\mathbf{0.29}_{\pm.10}$ | $\mathbf{0.17}_{\pm.36}$ |

$\hat{t}$ denote the ground-truth optimal dosage and the model's estimated dosage, respectively. Statistics are reported as the mean and standard error over multiple runs.

### 6.1. Synthetic Datasets

To evaluate our algorithm's dependency on Assumption 4.1 and its generalization capabilities, we constructed four synthetic datasets of increasing complexity. Hard Non-Linear (HNL) features a smooth, unimodal dose-response curve, serving as a baseline where assumptions hold perfectly. Complex Sharp Concave (CSC) increases difficulty with a highly non-linear neural network mapping for $t^*(x)$ and sharp response peaks, testing robustness under extreme curvature conditions. Simple Wavy (SWV) introduces multimodality that violates Assumption 4.1, evaluating the algorithm's global exploration capabilities when theoretical premises are compromised. And Smooth Tanh Saturation (STS) models a plateau-shaped OBD-style response with a flat optimal region, serving as an empirical testbed for Corollary 4.1 to access generalization beyond strict concavity and unique optima.

As illustrated in Table 1, GVALID consistently outperforms all baselines across the three synthetic datasets. On the HNL dataset, it exhibits dominant convergence speed and precision, significantly surpassing others under ideal theoretical conditions. In the CSC setting, while highly non-linear mappings and sharp response peaks increase performance variance, GVALID avoids the over-smoothing of bandit methods and the budget inefficiency of traditional active learning. By precisely maximizing gradient information gain, it maintains superior sample efficiency despite these curvature challenges. On the SWV dataset, although initial performance is limited by assumption violations, GVALID leverages the non-parametric flexibility of Gaussian Processes to rapidly correct model bias as samples accumulate. It successfully recovers from theoretical discrepancies to identify the global optimum, ultimately achieving superior asymptotic performance. Finally, on the STS dataset,

GVALID maintains reasonable generalization despite the violation of the unimodality assumption, likely because its gradient-based surrogate remains sensitive to local curvature compared with competing methods.

### 6.2. Semi-Synthetic Benchmark

To validate adaptability to realistic data distributions, we introduced the Semi-Synthetic News dataset (Schwab et al., 2020). Distinguished from synthetic baselines, this dataset leverages 500-dimensional features from real news logs to preserve authentic covariate sparsity and correlations. By constructing known reward functions over real feature distributions, this semi-synthetic approach allows us to evaluate generalization in high-dimensional scenarios with the precision of ground-truth availability.

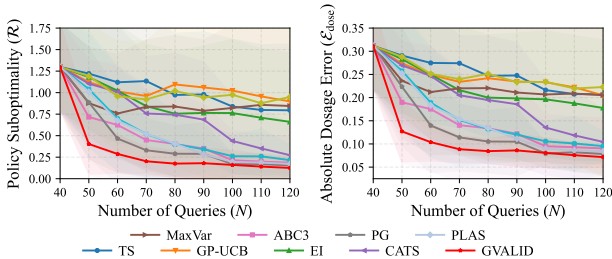

*Figure 2.* Performance on the News dataset. We report policy suboptimality ($\mathcal{R}$) and provide absolute dosage error ($\mathcal{E}_{\text{dose}}$) as a supplementary visualization of dosage estimation convergence.

Figure 2 illustrates the performance evolution on the News dataset. In addition to the primary metric $\mathcal{R}$, we introduce the absolute dosage error $\mathcal{E}_{\text{dose}} := \mathbb{E}[|\hat{t} - t^*|]$ as an auxiliary diagnostic. The synchronized decline of both metrics confirms that the reduction in suboptimality is directly driven by the increasing precision of optimal dosage identification, empirically validating the theoretical bound in Lemma 4.1 regarding their intrinsic relationship. Furthermore, the results underscore the challenge of robust estimation on high-dimensional sparse manifolds, where increased dimensionality and complexity lead to higher performance

volatility across all algorithms. While bandit methods struggle with inefficient confidence bounds and active learning baselines are prone to outlier distraction, GVALID excels by exploiting off-diagonal kernel information. By maximizing the covariance between candidates and target gradients, GVALID prioritizes samples strongly correlated with optimal dosage inference. This correlation-based mechanism effectively mitigates the limitations of high-dimensional distance metrics, enabling more targeted and efficient sampling on real-world distributions.

### 6.3. Ablation Study

To further ablate the efficacy of individual components within our method, this section compares the performance of GVALID against two variants across four datasets: GVALID-F, which replaces the gradient variance of the GP-estimated dose-response function with the function value itself as the sample selection criterion; and GVALID-RX, which omits the joint optimization of covariates $\mathbf{x}$ and dose $t$, instead randomly sampling $\mathbf{x}$ while optimizing only $t$.

As shown in Table 2, ablation results confirm that both the gradient-based uncertainty metric and the joint optimization strategy are critical to GVALID's performance. Removing or modifying either component leads to a significant increase in policy suboptimality, verifying the necessity of their synergistic operation. Notably, however, on the SWV dataset where Assumption 4.1 is violated, the advantage of GVALID over the random-sampling variant (GVALID-RX) narrows. We hypothesize that in scenarios where theoretical assumptions concerning smoothness do not hold, introducing randomness may facilitate an escape from local optima, thereby offering unexpected adaptability.

### 6.4. Hyperparameter Sensitivity

This section presents a sensitivity analysis regarding two active learning hyperparameters: the batch size per iteration and the ratio of initial samples to the total available pool. The sensitivity to the finite-difference step size $\delta$ is analyzed separately in Appendix H.3.

#### 6.4.1. BATCH SIZE

To investigate the trade-off between sample efficiency and computational cost, we evaluated batch sizes of $\{1, 5, 10, 15, 20\}$ under a fixed query budget. Figure 3 reveals the significant impact of batch size on algorithm performance: both extreme settings (e.g., batch sizes of 1 or 20) lead to increased variance and suboptimal results. Analysis suggests that excessively small batches may cause the Gaussian Process updates to be biased and unstable, trapping estimates in local optima. Conversely, larger batches are limited by the greedy strategy employed during candidate selection, where cumulative single-step errors degrade over-

all effectiveness. Balancing sample efficiency, stability, and computational cost, we conclude that a moderate batch size constitutes the optimal configuration.

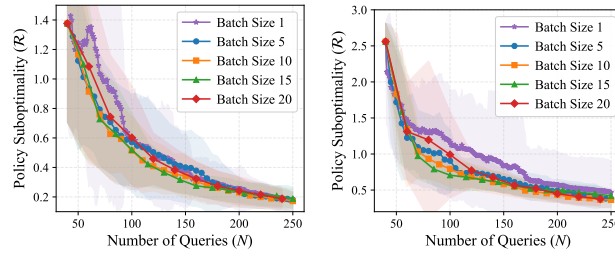

*(a)* Result on HNL Dataset     *(b)* Result on SWV Dataset

*Figure 3.* Impact of batch size on policy suboptimality ($\mathcal{R}$).

#### 6.4.2. INITIALIZATION RATIO

To investigate how the initial sample volume affects different methods, we performed a sensitivity analysis on the initial sample ratio ($\beta_{init}/\beta_{total}$) under a fixed total budget ($\beta_{total}$). We evaluated the impact of varying initialization ratios on each method's performance by recording the final policy suboptimality ($\mathcal{R}$) upon budget exhaustion.

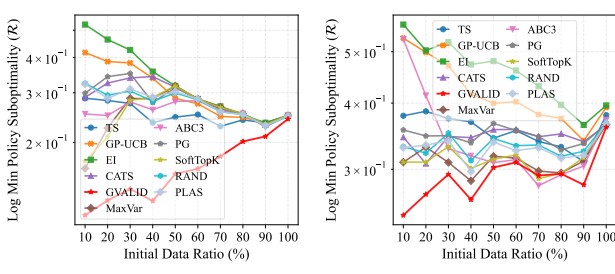

*(a)* Result on HNL Dataset     *(b)* Result on SWV Dataset

*Figure 4.* Impact of initial ratio on policy suboptimality ($\mathcal{R}$).

Figure 4 highlights significant disparities in initialization sensitivity among the evaluated methods. Some methods, such as EI and GP-UCB, exhibit a heavy reliance on the initial sample size; their performance degrades sharply under low initialization ratios (10%–20%), a trend particularly pronounced in the HNL task (Figure 4a). Notably, several baselines exhibit counterproductive sampling behaviors, where the performance after active selection is inferior to a configuration where the entire budget is allocated to random initialization. This phenomenon indicates that when the surrogate model is insufficiently trained due to limited initial samples, the selection mechanisms of these baselines are severely compromised. In contrast, GVALID maintains a substantial lead even with a minimal initial budget, demonstrating that its selection strategy remains effective despite the initial under-training of the surrogate model. This robustness highlights a superior cold-start advantage over other methods. Note that all results presented here utilize Latin Hypercube Sampling (LHS) for initialization; a detailed

*Table 2.* Ablation study of GVALID components during the convergence process ($N = 100$) and at steady-state convergence ($N = 300$).

| METHOD | $N = 100$ | | | | $N = 300$ | | | |
|---|---|---|---|---|---|---|---|---|
| | HNL | CSC | SWV | NEWS | HNL | CSC | SWV | NEWS |
| GVALID-RX | $0.68_{\pm.32}$ | $1.88_{\pm2.1}$ | $0.92_{\pm.53}$ | $0.18_{\pm.10}$ | $0.25_{\pm.10}$ | $1.09_{\pm2.5}$ | $0.35_{\pm.12}$ | $0.09_{\pm.05}$ |
| GVALID-F | $0.94_{\pm.43}$ | $1.73_{\pm1.4}$ | $0.93_{\pm.31}$ | $0.55_{\pm.85}$ | $0.26_{\pm.09}$ | $0.77_{\pm.39}$ | $0.37_{\pm.17}$ | $0.30_{\pm.66}$ |
| **GVALID** | $\mathbf{0.57_{\pm.33}}$ | $\mathbf{1.61_{\pm1.2}}$ | $\mathbf{0.91_{\pm.44}}$ | $\mathbf{0.16_{\pm.17}}$ | $\mathbf{0.15_{\pm.06}}$ | $\mathbf{0.48_{\pm.31}}$ | $\mathbf{0.33_{\pm.10}}$ | $\mathbf{0.07_{\pm.03}}$ |

discussion on the advantages of LHS over Random Sampling (RAND), along with comprehensive results for other datasets, is provided in Appendix H.2.

### 6.5. Empirical Validation of Theorem

This experiment aims to validate the consistency between our theoretical regret bound and empirical algorithm performance. As shown in Figure 5, the theoretical upper bound derived from posterior variance (red line) effectively tracks the descending trajectory of policy suboptimality (blue line), providing numerical validation for the derivations in Theorem 4.1. Specifically, the coefficient $C_g$ within the theoretical bound is computed based on our estimates of the regularity parameters $\mu$ and $L$; the specific values of these estimates for each dataset are provided in Appendix F.6.

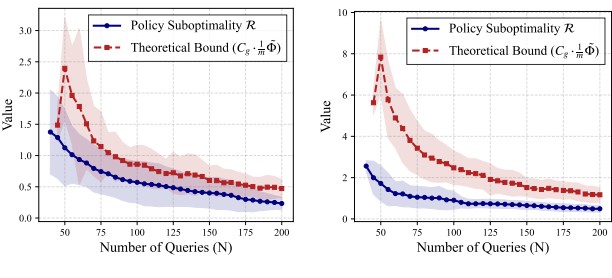

*(a)* Result on HNL Dataset    *(b)* Result on SWV Dataset

*Figure 5.* Validation of the theoretical regret bound against empirical performance.

## 7. Conclusion

We recast budget-constrained individualized dosing as a policy optimization problem, proving that regret is bounded by the posterior gradient variance at estimated optima. This perspective informs GVALID, a batched active learning algorithm that utilizes finite-difference proxies and greedy updates to prioritize gradient uncertainty reduction. Theoretical and empirical results confirm that this gradient-aware strategy enhances sample efficiency for policy learning under restricted experimental budgets.

## Acknowledgments

This research was supported in part by the National Nature Science Foundation of China (No. 62137002), in part by the USTC Research Funds of the Double First-Class Initiative (Grant No. YD9110002085), in part by the Research Funds of Centre for Leading Medicine and Advanced Technologies of IHM (Grant No. 2025IHM01030).

## Impact Statement

This paper advances methods for budget-constrained individualized dosing, aiming to make experimental design more sample-efficient and reduce unnecessary interventions. When applied to high-stakes domains such as healthcare, our framework should be accompanied by careful consideration of fairness, transparency, and appropriate clinical oversight. Beyond these standard concerns for decision-making systems in sensitive settings, we do not anticipate additional societal risks specific to our method.

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

# A. Extended Related Work

This appendix provides a detailed review of the literature that complements the main text. We detail the methodological diversity in active learning for causal inference, the theoretical evolution of batched contextual bandits, decision-focused perspectives from best-arm identification, and parallel work in clinical dose finding.

## A.1. Extended Review of Active Learning

Broadly, active learning (Cohn et al., 1996) focuses on adaptively selecting informative samples under budget constraints. In general batch query scenarios, significant research has optimized sampling strategies: Gal et al. (2017) utilized information gain for query selection in Gaussian Process regression, while Kirsch et al. (2019) extended Bayesian methods to deep learning frameworks to handle high-dimensional settings.

In the causal domain, methodologies initially diverged toward observational data with label budgets, where approaches such as (Jesson et al., 2021; Wen et al., 2025) leverage coverage maximization theory to accurately estimate effects while minimizing labeling costs. The second category of causal work, which aligns with our experimental focus, involves active intervention. For binary treatments, Puha et al. (2020) combined Bayesian Additive Regression Trees (BART) with uncertainty maximization. Similarly, Cha & Lee (2025) adopted the Cohn criterion to reduce the estimation error of conditional treatment effects by minimizing the integrated posterior variance, while others employed leverage score sampling (Addanki et al., 2022; Ghadiri et al., 2023; Zhang et al., 2025) to optimize estimation. For continuous treatments, Toth et al. (2022) modeled variable relationships using Gaussian processes under nonlinear additive noise models, selecting data by maximizing mutual information. Although originally developed for bandit settings, the framework by Razzak et al. (2024) is regarded here as a contribution to batch active learning due to its mechanism for actively selecting covariates from a candidate pool to optimize batch expected information gain. Despite these methodological advances, these works primarily aim to minimize *estimation error*, distinguishing them from our approach which targets direct policy value maximization.

## A.2. Extended Review of Batched Contextual Bandits

BCB extends the standard framework by introducing batch constraints where decisions are simultaneous and feedback is delayed. Early theoretical foundations were laid by Gao et al. (2019) for multi-armed bandits. In the contextual setting, linear models have been extensively studied using action elimination (Esfandiari et al., 2021) or optimization-based design (Hanna et al., 2023), extending to Generalized Linear Models (GLMs) (Ren et al., 2020; Kalkanli & Ozgur, 2021) and adaptive batch sizes (Sawarni et al., 2024). To address complex reward structures beyond parametric limits, recent research has explored nonparametric settings. For context-free scenarios, (Kathuria et al., 2016; Li & Scarlett, 2022) utilized Gaussian Process rewards, while in contextual settings, Feng et al. (2022) addressed metric action spaces under Lipschitz assumptions via suboptimal arm elimination. Similarly, Jiang & Ma (2025) investigated context-smooth bandits using adaptive context partitioning, though this was restricted to binary actions.

In parallel, applied research in medical and statistical literature (Villar et al., 2015; Villar & Rosenberger, 2018; Varatharajah & Berry, 2022) has leveraged BCB for treatment policy optimization. These studies typically implement Upper Confidence Bound (UCB) or Thompson sampling methods within classical bandit frameworks, primarily focusing on finite discrete action sets without incorporating the active covariate selection mechanisms that are central to our work.

## A.3. Best-Arm Identification.

Orthogonal to the batching constraint discussed in batched contextual bandits, our objective is closely related to best-arm identification, since the goal is not to maximize cumulative reward during data collection but to output a high-value policy after the experiment. In this literature, performance is evaluated by final decision quality, typically measured by simple regret or the probability of incorrect selection (Bubeck et al., 2011; Kaufmann et al., 2016). Bubeck et al. (2011) formalized simple regret as a criterion for pure exploration in finite- and continuous-armed bandits, while Kaufmann et al. (2016) characterized instance-dependent complexity in both fixed-budget and fixed-confidence best-arm identification settings. Subsequent extensions to structured decision spaces further study optimal sample allocation and information-theoretic lower bounds for final decision accuracy, including transductive and linear bandits (Fiez et al., 2019; Jedra & Proutiere, 2020). More directly related to our setting, Kato et al. (2024) formulates adaptive experimental design for policy learning as contextual BAI and proposes PLAS with matching lower- and upper-bound guaranties. These works provide an important decision-focused lens for our problem. However, these works primarily consider finite, discrete, or linearly parameterized arms, whereas

our setting involves individualized continuous dosing under one-shot batched experimentation. Our contribution can therefore be understood as adapting this decision-focused principle to continuous-dose policy learning by exploiting smooth dose-response geometry through a tractable gradient-variance surrogate.

## A.4. Dose Finding

Another line of research relevant to our context is dose finding, where the goal is to identify the maximum tolerated dose or the most successful dose using limited samples. Bayesian adaptive designs are widely employed in this setting, typically relying on monotonic toxicity assumptions and utilizing sequential prior modeling with cohort-level updates to optimize dose selection (O'Quigley & Conaway, 2010). Recently, these methods have been analyzed through a multi-armed bandit lens, where researchers have bridged clinical design and learning theory by establishing sample complexity and regret guarantees (Aziz et al., 2021). However, whereas these works focus on identifying a global optimal dose under toxicity constraints, they generally overlook individual heterogeneity. In contrast, our work adopts an individualized dose response framework.

## B. Assumptions

**Assumption B.1** (Consistency). If an individual receives dose $t$, then the observed outcome equals the corresponding potential outcome: $Y = Y^t$.

**Assumption B.2** (Ignorability). Potential outcomes are independent of the assigned dose conditional on covariates:

$$\{Y_x^t\}_{t \in \mathcal{T}} \perp T \mid X. \tag{20}$$

**Assumption B.3** (Positivity). All doses have positive assignment probability density for every covariate profile:

$$\exists c > 0 \text{ s.t. } \mathbb{P}(T = t \mid X = x) \geq c, \forall x \in \mathcal{X}, \ t \in \mathcal{T}. \tag{21}$$

Our algorithm is designed to strictly adhere to these assumptions. Specifically, the use of a GP to model observed outcomes aligns with Assumption B.1. Since the acquisition score $\Delta_{\mathcal{D}^\tau}^{\mathcal{S}}(q)$ depends solely on the posterior variance, which is independent of the observed outcome values $y$, the query selection remains independent of potential outcomes, thereby satisfying Assumption B.2. Furthermore, by stochastically sampling dosages within each grid cell $I_k$ as defined in Equation (14), the algorithm ensures non-zero coverage across the entire intervention domain $\mathcal{T}$, satisfying Assumption B.3.

**Assumption B.4** (Interior Posterior-Mean Maximizer). There exists $\delta \in (0, 1/2)$ such that the posterior mean $\mu_{\mathcal{D}}(x, t) := \mathbb{E}[f(x, t) \mid \mathcal{D}]$ is differentiable in $t$ on $T_\delta := [\delta, 1 - \delta]$, and the policy $\hat{t}_{\mathcal{D}}^*(x)$ is selected as an interior maximizer:

$$\hat{t}_{\mathcal{D}}^*(x) \in \arg\max_{t \in T_\delta} \mu_{\mathcal{D}}(x, t) \quad \text{and} \quad \hat{t}_{\mathcal{D}}^*(x) \in (\delta, 1 - \delta), \ \forall x \in \mathcal{X}. \tag{22}$$

Assumption B.4 ensures that the estimated optimal dose $\hat{t}_{\mathcal{D}}^*(\mathbf{x})$ is a stationary point of the posterior mean $\mu_{\mathcal{D}}(\mathbf{x}, t)$, implying the vanishing of the expected gradient $\mathbb{E}[g(\mathbf{x}, \hat{t}_{\mathcal{D}}^*(\mathbf{x})) \mid \mathcal{D}] = 0$. This interior-optimum condition is practically reasonable in dose-response studies, particularly in modern dose-optimization trials that aim to identify an optimal biological dose balancing efficacy and safety, rather than simply selecting the maximum tolerated or boundary dose (Yuan et al., 2024).

## C. Supporting Theoretical Results

**Theorem C.1** (Finite-sample Bound). *Under Assumption 4.1, at any round $\tau$, let $\mathbf{x}_1, \ldots, \mathbf{x}_m \overset{\text{i.i.d.}}{\sim} P_X$ and construct $\mathcal{Z}^\tau$ accordingly. Then, for any $\delta \in (0, 1)$, with probability at least $1 - \delta$, we have*

$$R(\mathcal{D}^\tau) \leq C_{\mathrm{g}} \left[ \frac{1}{m} \Phi_{\mathcal{D}^\tau}(\mathcal{Z}^\tau) + G_m^2 \sqrt{\frac{\log(2/\delta)}{2m}} \right], \tag{23}$$

*where $\Phi_{\mathcal{D}^\tau}(\mathcal{Z}^\tau)$ is defined in Equation (10).*

## D. Proof

### D.1. Proof of Lemma 4.1

*Proof.* Fix $\mathbf{x} \in \mathcal{X}$ and let $t^* = t^*(\mathbf{x})$. By definition of $t^*$ as a maximizer and assuming it lies in the interior of $\mathcal{T}$, the first-order optimality condition gives:

$$\partial_t f(\mathbf{x}, t^*) = 0. \tag{24}$$

For any $t \in \mathcal{T}$, Taylor's theorem with Lagrange remainder (applicable since $f$ is twice times continuously differentiable by Assumption 4.1) guarantees the existence of $\xi$ between $t$ and $t^*$ such that:

$$f(\mathbf{x}, t) = f(\mathbf{x}, t^*) + \partial_t f(\mathbf{x}, t^*)(t - t^*) + \frac{1}{2}\partial_{tt} f(\mathbf{x}, \xi)(t - t^*)^2 \tag{25}$$

$$= f(\mathbf{x}, t^*) + \frac{1}{2}\partial_{tt} f(\mathbf{x}, \xi)(t - t^*)^2, \tag{26}$$

where the second equation follows from Equation (24).

The pointwise regret is therefore:

$$R(\mathbf{x}; t) = f(\mathbf{x}, t^*) - f(\mathbf{x}, t) = -\frac{1}{2}\partial_{tt} f(\mathbf{x}, \xi)(t - t^*)^2. \tag{27}$$

From Assumption 4.1 1), we have the bound $-\partial_{tt} f(\mathbf{x}, t) \leq L$ for all $(\mathbf{x}, t) \in \mathcal{X} \times \mathcal{T}$. Applying this to $\xi$ yields:

$$-\partial_{tt} f(\mathbf{x}, \xi) \leq L. \tag{28}$$

Multiplying by the non-negative quantity $\frac{1}{2}(t - t^*)^2$ preserves the inequalities:

$$-\frac{1}{2}\partial_{tt} f(\mathbf{x}, \xi)(t - t^*)^2 \leq \frac{L}{2}(t - t^*)^2. \tag{29}$$

The proof is thus completed. Note that if $t^*$ lies on the boundary of $\mathcal{T}$, the first-order condition may not hold, but the quadratic bounds remain valid by continuity of $\partial_{tt} f$ and the fact that regret is minimized at the boundary optimum. $\square$

### D.2. Proof of Theorem 4.1

*Proof.* By Equation (7) and the upper bound in Lemma 4.1, we have

$$R(\mathcal{D}) = \mathbb{E}_X\left[\mathbb{E}\left[R(X; \hat{t}^*_{\mathcal{D}}(X)) \mid \mathcal{D}\right]\right] \leq \frac{L}{2}\mathbb{E}_X\left[\mathbb{E}\left[|t^*(X) - \hat{t}^*_{\mathcal{D}}(X)|^2 \mid \mathcal{D}\right]\right]. \tag{30}$$

By definition of the true optimum, $g(\mathbf{x}, t^*(\mathbf{x})) = \partial_t f(\mathbf{x}, t)\big|_{t=t^*(\mathbf{x})} = 0$. Apply the Mean Value Theorem to $g(\mathbf{x}, \cdot)$ between $t^*(\mathbf{x})$ and $\hat{t}^*_{\mathcal{D}}(\mathbf{x})$: there exists $\zeta = (1 - \lambda)t^*(\mathbf{x}) + \lambda\hat{t}^*_{\mathcal{D}}(\mathbf{x})$ for some $\lambda \in (0, 1)$ such that:

$$0 - g(\mathbf{x}, \hat{t}^*_{\mathcal{D}}(\mathbf{x})) = \partial_t g(\mathbf{x}, \zeta)(t^*(\mathbf{x}) - \hat{t}^*_{\mathcal{D}}(\mathbf{x}))$$

$$= \partial_{tt} f(\mathbf{x}, \zeta)(t^*(\mathbf{x}) - \hat{t}^*_{\mathcal{D}}(\mathbf{x})). \tag{31}$$

By Assumption 4.1 1), $-\partial_{tt} f(\mathbf{x}, \zeta) \geq \mu > 0$, so we have

$$|t^*(\mathbf{x}) - \hat{t}^*_{\mathcal{D}}(\mathbf{x})| \leq \frac{|g(\mathbf{x}, \hat{t}^*_{\mathcal{D}}(\mathbf{x}))|}{\mu}. \tag{32}$$

Squaring both sides and taking the expectation conditioned on the observed data $\mathcal{D}$, we obtain:

$$\mathbb{E}\left[|t^*(\mathbf{x}) - \hat{t}^*_{\mathcal{D}}(\mathbf{x})|^2 \mid \mathcal{D}\right] \leq \frac{1}{\mu^2}\mathbb{E}\left[g(\mathbf{x}, \hat{t}^*_{\mathcal{D}}(\mathbf{x}))^2 \mid \mathcal{D}\right] \tag{33}$$

$$= \frac{1}{\mu^2}\left(\left(\mathbb{E}[g(\mathbf{x}, \hat{t}^*_{\mathcal{D}}(\mathbf{x})) \mid \mathcal{D}]\right)^2 + \mathrm{Var}(g(\mathbf{x}, \hat{t}^*_{\mathcal{D}}(\mathbf{x})) \mid \mathcal{D})\right), \tag{34}$$

$$\leq \frac{1}{\mu^2}\mathrm{Var}(g(\mathbf{x}, \hat{t}^*_{\mathcal{D}}(\mathbf{x})) \mid \mathcal{D}), \tag{35}$$

where Equation (34) follows from the identity $\mathbb{E}[X^2] = (\mathbb{E}[X])^2 + \mathrm{Var}(X)$, and Equation (35) is obtained by invoking Assumptions B.4 and 4.1, which implies that at the stationary point, the squared expected gradient is zero.

Taking the expectation over the covariates $X$ and substituting Equation (35) into Equation (30) yields:

$$R(\mathcal{D}) \leq \frac{L}{2\mu^2} \mathbb{E}_X \left[ \mathrm{Var}\big(g(X, \hat{t}^*_{\mathcal{D}}(X)) | \mathcal{D}\big) \right]. \tag{36}$$

$\square$

### D.3. Proof of Corollary 4.1

*Proof.* For each $\mathbf{x}$, since $f(\mathbf{x}, t)$ is constant on $\mathcal{T}^\star(\mathbf{x})$, the choice of $t^\star \in \mathcal{T}^\star(\mathbf{x})$ does not affect the value of $f(\mathbf{x}, t^\star)$. Therefore, the posterior-conditional regret of the posterior-mean policy can be written as

$$R(\mathcal{D}) = \mathbb{E}_X \left[ \mathbb{E}\big[ f(X, t^\star) - f(X, \hat{t}^*_{\mathcal{D}}(X)) \mid \mathcal{D} \big] \right], \tag{37}$$

where $t^\star \in \mathcal{T}^\star(X)$.

Applying the set-valued regret–gradient condition with $t = \hat{t}^*_{\mathcal{D}}(X)$ gives

$$f(X, t^\star) - f(X, \hat{t}^*_{\mathcal{D}}(X)) \leq \frac{L}{2\mu^2} \left| g(X, \hat{t}^*_{\mathcal{D}}(X)) \right|^2. \tag{38}$$

Taking conditional expectation given $\mathcal{D}$ and then expectation over $X$, we obtain

$$R(\mathcal{D}) \leq \frac{L}{2\mu^2} \mathbb{E}_X \left[ \mathbb{E}\Big[ \left| g(X, \hat{t}^*_{\mathcal{D}}(X)) \right|^2 \mid \mathcal{D} \Big] \right]. \tag{39}$$

By Assumption 4.2, $g(\mathbf{x}, t)$ is a Gaussian process. Moreover, since $\hat{t}^*_{\mathcal{D}}(\mathbf{x})$ maximizes the posterior mean and satisfies the same interior first-order condition used in Theorem 4.1, we have

$$\mathbb{E}\big[ g(X, \hat{t}^*_{\mathcal{D}}(X)) \mid \mathcal{D} \big] = \partial_t \mu_{\mathcal{D}}(X, \hat{t}^*_{\mathcal{D}}(X)) = 0. \tag{40}$$

Hence its conditional second moment equals its conditional variance:

$$\mathbb{E}\Big[ \left| g(X, \hat{t}^*_{\mathcal{D}}(X)) \right|^2 \mid \mathcal{D} \Big] = \mathrm{Var}\big( g(X, \hat{t}^*_{\mathcal{D}}(X)) \mid \mathcal{D} \big). \tag{41}$$

Substituting this identity yields

$$R(\mathcal{D}) \leq \frac{L}{2\mu^2} \mathbb{E}_X \left[ \mathrm{Var}\big( g(X, \hat{t}^*_{\mathcal{D}}(X)) \mid \mathcal{D} \big) \right], \tag{42}$$

which is the same regret bound as in Theorem 4.1. Thus, the bound remains valid after replacing the singleton optimum with the flat optimal set $\mathcal{T}^\star(\mathbf{x})$. $\square$

### D.4. Proof of Theorem 5.1

*Proof.* We analyze the reduction in posterior uncertainty given the fixed historical dataset $\mathcal{D}^\tau$ and the currently selected batch subset $\mathcal{S}$. We structure the proof by first establishing the update rule for the finite-difference vector $\mathbf{f}_{\mathrm{fd}}$ upon observing a new candidate $q$, and then propagating this update to the gradient proxy $\tilde{\mathbf{g}}$.

**Step 1. Linear Representation.** Recall from Section 5.1 that the finite-difference vector is defined as $\mathbf{f}_{\mathrm{fd}} \in \mathbb{R}^{2m}$, where $m$ is the size of the target set. As established in Equation (12), the gradient proxy $\tilde{\mathbf{g}} \in \mathbb{R}^m$ is a fixed linear transformation of $\mathbf{f}_{\mathrm{fd}}$. Thus, there exists a sparse matrix $\mathbf{A} \in \mathbb{R}^{m \times 2m}$ such that $\tilde{\mathbf{g}} = \mathbf{A} \mathbf{f}_{\mathrm{fd}}$.

**Step 2. Joint Distribution and Covariance Update.** Consider the joint distribution of $\mathbf{f}_{\text{fd}}$ and the candidate observation $y_q = f(q) + \varepsilon$. Under the GP prior (Assumption 4.2), conditioned on the information set $\mathcal{D}^\tau \cup \mathcal{S}$, this joint vector follows a multivariate Gaussian distribution. We partition the conditional covariance matrix as follows:

$$\text{Cov}\left(\begin{bmatrix} \mathbf{f}_{\text{fd}} \\ y_q \end{bmatrix} \,\Big|\, \mathcal{D}^\tau, \mathcal{S}\right) = \begin{bmatrix} \Sigma_{\text{fd}}^{\mathcal{S}} & \mathbf{k}_{\text{fd},q}^{\mathcal{S}} \\ (\mathbf{k}_{\text{fd},q}^{\mathcal{S}})^\top & s_{\mathcal{D}^\tau}^{\mathcal{S}}(q) \end{bmatrix} \in \mathbb{R}^{(2m+1)\times(2m+1)}, \tag{43}$$

where the notation explicitly denotes dependence on the current subset $\mathcal{S}$:

- $\Sigma_{\text{fd}}^{\mathcal{S}} := \text{Cov}(\mathbf{f}_{\text{fd}} \mid \mathcal{D}^\tau, \mathcal{S}) \in \mathbb{R}^{2m \times 2m}$ is the current covariance of the finite-difference values;

- $\mathbf{k}_{\text{fd},q}^{\mathcal{S}} := \text{Cov}(\mathbf{f}_{\text{fd}}, y_q \mid \mathcal{D}^\tau, \mathcal{S}) \in \mathbb{R}^{2m}$ is the cross-covariance vector;

- $s_{\mathcal{D}^\tau}^{\mathcal{S}}(q) := \text{Var}(y_q \mid \mathcal{D}^\tau, \mathcal{S}) \in \mathbb{R}$ is the predictive variance of the candidate (including noise), matching the definition in Theorem 5.1.

Upon observing $y_q$, the information set expands to $\mathcal{D}^\tau \cup \mathcal{S} \cup \{q\}$. The covariance of $\mathbf{f}_{\text{fd}}$ is updated via the standard Schur complement:

$$\Sigma_{\text{fd}}^{\mathcal{S}\cup\{q\}} = \Sigma_{\text{fd}}^{\mathcal{S}} - \frac{\mathbf{k}_{\text{fd},q}^{\mathcal{S}}(\mathbf{k}_{\text{fd},q}^{\mathcal{S}})^\top}{s_{\mathcal{D}^\tau}^{\mathcal{S}}(q)}. \tag{44}$$

**Step 3. Propagation to the Gradient Proxy.** We now derive the update rule for the gradient proxy covariance, denoted by $\Sigma_{\tilde{g}}^{\mathcal{S}} := \text{Cov}(\tilde{\mathbf{g}} \mid \mathcal{D}^\tau, \mathcal{S}) \in \mathbb{R}^{m \times m}$. By the linearity of the covariance operator and the relation $\tilde{\mathbf{g}} = \mathbf{A}\mathbf{f}_{\text{fd}}$, the updated covariance is:

$$\Sigma_{\tilde{g}}^{\mathcal{S}\cup\{q\}} = \mathbf{A}\Sigma_{\text{fd}}^{\mathcal{S}\cup\{q\}}\mathbf{A}^\top. \tag{45}$$

Substituting the finite-difference update (44) into this expression yields:

$$\begin{aligned}
\Sigma_{\tilde{g}}^{\mathcal{S}\cup\{q\}} &= \mathbf{A}\left(\Sigma_{\text{fd}}^{\mathcal{S}} - \frac{\mathbf{k}_{\text{fd},q}^{\mathcal{S}}(\mathbf{k}_{\text{fd},q}^{\mathcal{S}})^\top}{s_{\mathcal{D}^\tau}^{\mathcal{S}}(q)}\right)\mathbf{A}^\top \\
&= \underbrace{\mathbf{A}\Sigma_{\text{fd}}^{\mathcal{S}}\mathbf{A}^\top}_{\Sigma_{\tilde{g}}^{\mathcal{S}}} - \frac{(\mathbf{A}\mathbf{k}_{\text{fd},q}^{\mathcal{S}})(\mathbf{A}\mathbf{k}_{\text{fd},q}^{\mathcal{S}})^\top}{s_{\mathcal{D}^\tau}^{\mathcal{S}}(q)}.
\end{aligned} \tag{46}$$

Observe that the numerator term involves the linear projection of the cross-covariance vector. This term precisely recovers the covariance between the gradient proxy and the observation:

$$\mathbf{A}\mathbf{k}_{\text{fd},q}^{\mathcal{S}} = \text{Cov}(\mathbf{A}\mathbf{f}_{\text{fd}}, y_q \mid \mathcal{D}^\tau, \mathcal{S}) = \mathbf{c}_{\mathcal{D}^\tau}^{\mathcal{S}}(q) \in \mathbb{R}^m, \tag{47}$$

where $\mathbf{c}_{\mathcal{D}^\tau}^{\mathcal{S}}(q)$ is defined in Theorem 5.1. Thus, the update simplifies to:

$$\Sigma_{\tilde{g}}^{\mathcal{S}\cup\{q\}} = \Sigma_{\tilde{g}}^{\mathcal{S}} - \frac{\mathbf{c}_{\mathcal{D}^\tau}^{\mathcal{S}}(q)\mathbf{c}_{\mathcal{D}^\tau}^{\mathcal{S}}(q)^\top}{s_{\mathcal{D}^\tau}^{\mathcal{S}}(q)}. \tag{48}$$

**Step 4. Trace Reduction.** The objective reduction $\Delta_{\mathcal{D}^\tau}^{\mathcal{S}}(q)$ is defined as the difference in the traces of the covariance matrices before and after adding $q$. Applying the trace operator to the derived update:

$$\Delta_{\mathcal{D}^\tau}^{\mathcal{S}}(q) := \text{tr}(\Sigma_{\tilde{g}}^{\mathcal{S}}) - \text{tr}(\Sigma_{\tilde{g}}^{\mathcal{S}\cup\{q\}}) = \text{tr}\left(\frac{\mathbf{c}_{\mathcal{D}^\tau}^{\mathcal{S}}(q)\mathbf{c}_{\mathcal{D}^\tau}^{\mathcal{S}}(q)^\top}{s_{\mathcal{D}^\tau}^{\mathcal{S}}(q)}\right). \tag{49}$$

Using the cyclic property of the trace ($\text{tr}(\mathbf{u}\mathbf{u}^\top) = \|\mathbf{u}\|_2^2$ for any vector $\mathbf{u}$), we recover the result in Equation (16):

$$\Delta_{\mathcal{D}^\tau}^{\mathcal{S}}(q) = \frac{\left\|\mathbf{c}_{\mathcal{D}^\tau}^{\mathcal{S}}(q)\right\|_2^2}{s_{\mathcal{D}^\tau}^{\mathcal{S}}(q)}. \tag{50}$$

$\square$

### D.5. Proof of Theorem C.1

*Proof.* Fix the current dataset $\mathcal{D}^\tau$ and define the estimated optimal dose $t_\tau(\mathbf{x}) := \arg\max_{t \in \mathcal{T}} \mu_{\mathcal{D}^\tau}(\mathbf{x}, t)$. Let $\psi_{\mathcal{D}^\tau}(\mathbf{x}) := \mathrm{Var}\big(g(\mathbf{x}, t_\tau(\mathbf{x})) \mid \mathcal{D}^\tau\big)$ denote the posterior variance of the gradient. By Theorem 4.1, the cumulative Bayes regret is bounded by the expected gradient variance:

$$R(\mathcal{D}^\tau) \leq C_g \, \mathbb{E}_{\mathbf{x} \sim P_X}\big[\psi_{\mathcal{D}^\tau}(\mathbf{x})\big]. \tag{51}$$

To approximate this expectation, our acquisition function $\Phi_{\mathcal{D}^\tau}$ is computed over the finite set $\mathcal{Z}^\tau = \{\mathbf{x}_j\}_{j=1}^m$:

$$\frac{1}{m}\Phi_{\mathcal{D}^\tau}(\mathcal{Z}^\tau) = \frac{1}{m}\sum_{j=1}^m \psi_{\mathcal{D}^\tau}(\mathbf{x}_j). \tag{52}$$

Under Assumption 4.1, the gradient $g$ is bounded by $G_m$, which implies that the variance satisfies $0 \leq \psi_{\mathcal{D}^\tau}(\mathbf{x}) \leq G_m^2$. Assuming the samples $\{\mathbf{x}_j\}_{j=1}^m$ are drawn i.i.d. from $P_X$, Hoeffding's inequality guarantees that for any $\delta \in (0, 1)$, with probability at least $1 - \delta$:

$$\mathbb{E}_{\mathbf{x} \sim P_X}\big[\psi_{\mathcal{D}^\tau}(\mathbf{x})\big] \leq \frac{1}{m}\sum_{j=1}^m \psi_{\mathcal{D}^\tau}(\mathbf{x}_j) + G_m^2 \sqrt{\frac{\log(1/\delta)}{2m}}. \tag{53}$$

Combining (51), (52), and (53), we obtain:

$$R(\mathcal{D}^\tau) \leq C_g \left[ \frac{1}{m}\Phi_{\mathcal{D}^\tau}(\mathcal{Z}^\tau) + G_m^2 \sqrt{\frac{\log(1/\delta)}{2m}} \right]. \tag{54}$$

This completes the proof, showing that minimizing the empirical acquisition function $\Phi_{\mathcal{D}^\tau}$ effectively minimizes the upper bound of the Bayes regret. □

## E. Detailed Algorithmic Framework

Algorithm 2 summarizes our batched active learning procedure for learning an ITR from a finite intervention pool. Starting from an initial random design of size $N_{\mathrm{init}}$, we fit a GP surrogate to the collected triplets $(\mathbf{x}, t, y)$ and maintain an available set of yet-unqueried individuals. At each outer round $\tau$, we invoke GVALID (Algorithm 1) to select a batch $\mathcal{S}_\tau$ of $B$ intervention queries based on the current GP posterior, query the interventional oracle for the corresponding outcomes, and update the dataset and GP posterior. After $T$ rounds (or once the pool is exhausted), we extract the final policy by maximizing the GP posterior mean over the dose grid.

## F. Datasets Description

In this section, we provide a detailed description of the synthetic and semi-synthetic datasets used in our evaluation. Figure 6 illustrates the dose-response curves for a set of randomly selected individuals across different environments, highlighting the heterogeneity in their response profiles.

The construction of four synthetic environments follows established benchmarks and dose-response patterns in the causal inference and dose-optimization literature. HNL and CSC are adapted from the continuous treatment simulations in (Bica et al., 2020; Kallus & Zhou, 2018), utilizing strong non-linearity and asymmetry to model heterogeneous treatment effects in clinical decision-making. SWV draws on multimodal optimization benchmarks to evaluate robustness against local optima. STS follows plateau-shaped dose-response patterns commonly discussed in the OBD literature, where efficacy saturates after a certain dose and further escalation may bring limited benefit but increased toxicity risk (Guo & Yuan, 2023). Collectively, these datasets encompass a range of realistic response profiles, from smooth unimodal to oscillatory forms, providing a rigorous validation of sample efficiency across diverse regularity regimes.

### F.1. Hard Non-Linear Dataset (HNL)

The HNL dataset models a challenging continuous dose-response setting in an 8-dimensional covariate space. Covariates $\mathbf{x} \in \mathbb{R}^8$ are sampled independently from a uniform distribution: $\mathbf{x} \sim \mathrm{Unif}(-1, 1)^8$. The outcome is generated from a

---

**Algorithm 2** Batched Active Learning of ITR

---

**Input:** pool $\mathcal{X}_{\text{pool}}$, interventional oracle $\mathcal{O}(\cdot, \cdot)$, rounds $T$, batch size $B$, initial budget $N_{\text{init}}$, GP prior, grid $\mathcal{T}_c$.
**Output:** learned ITR $\hat{\pi}(\cdot)$.
Initialize dataset $\mathcal{D}^0$ and available set $\mathcal{X}_{\text{avail}} \leftarrow \mathcal{X}_{\text{pool}}$.
**Initialization:** Select $N_{\text{init}}$ distinct individuals $\mathbf{x}_i$ from $\mathcal{X}_{\text{avail}}$ and assign initial doses $t_i$ randomly.
**for** each selected $(\mathbf{x}_i, t_i)$ **do**
    Query oracle $y_i \leftarrow \mathcal{O}(\mathbf{x}_i, t_i)$.
    Set $\mathcal{D}^0 \leftarrow \mathcal{D}^0 \cup \{(\mathbf{x}_i, t_i, y_i)\}$.
    Remove $\mathbf{x}_i$ from $\mathcal{X}_{\text{avail}}$.
**end for**
Fit GP posterior $(\mu^0, k^0)$ using $\mathcal{D}^0$.
**for** $\tau = 1$ **to** $T$ **do**
    **if** $\mathcal{X}_{\text{avail}}$ is empty **then**
        **break**
    **end if**
    Denote the GP posterior by $(\mu^\tau, k^\tau)$.
    Select $\mathcal{S}_\tau \leftarrow \text{GVALID}(\tau, (\mu^\tau, k^\tau), \mathcal{X}_{\text{pool}}, \mathcal{X}_{\text{avail}}, B, \mathcal{T}_c)$               (cf. Algorithm 1).
    Update $\mathcal{D}^\tau \leftarrow \mathcal{D}^{\tau-1}$.
    **for** each $(\mathbf{x}_b, t_b) \in S_\tau$ **do**
        $y_b \leftarrow \mathcal{O}(\mathbf{x}_b, t_b)$.
        Set $\mathcal{D}^\tau \leftarrow \mathcal{D}^\tau \cup \{(\mathbf{x}_b, t_b, y_b)\}$.
        Remove $\mathbf{x}_b$ from $\mathcal{X}_{\text{avail}}$.
    **end for**
    Update GP posterior $(\mu^{\tau+1}, k^{\tau+1})$ using $\mathcal{D}^\tau$.
**end for**
**Policy extraction:** $\hat{\pi}(\cdot) \leftarrow \arg\max_{t \in \mathcal{T}} \mu^T(\cdot, t)$.

---

concave response surface centered at an $\mathbf{x}$-dependent optimal dose $t^*(\mathbf{x})$, with a curvature $k(\mathbf{x})$ that varies with the position in the input space. Formally, the noiseless response function is defined as:

$$f(\mathbf{x}, t) = 10.0 - k(\mathbf{x}) \left( \cosh\left(t - t^*(\mathbf{x})\right) - 1 \right).$$

And the observed outcome $y$ is sampled according to Equation (1), and the noise $\sigma_\epsilon$ is set to 0.1. The optimal dose $t^*(\mathbf{x})$ is determined by a sinusoidal projection of the covariates followed by a sigmoid transformation:

$$t^*(\mathbf{x}) = 0.1 + 0.7 \cdot \sigma\left(5\sin(\mathbf{x}^\top \mathbf{w}_{\text{center}})\right),$$

where $\sigma(z) = (1 + e^{-z})^{-1}$ is the sigmoid function, and $\mathbf{w}_{\text{center}}$ is a fixed weight vector normalized to unit length ($\|\mathbf{w}_{\text{center}}\|_2 = 1$). The curvature term $k(\mathbf{x})$ increases the steepness of the response surface based on the Euclidean norm of the covariates:

$$k(\mathbf{x}) = 20.0 + 100 \cdot \max\left(0, \min(\|\mathbf{x}\|_2 - 1, 1)\right),$$

which simplifies to a clipped linear penalty when $\|\mathbf{x}\|_2 > 1$. This construction ensures that the response surface becomes substantially sharper as $\mathbf{x}$ moves away from the origin, making precise localization of $t^*(\mathbf{x})$ increasingly critical for regret minimization.

### F.2. Complex Sharp Concave Dataset (CSC)

The CSC dataset is an 8-dimensional synthetic benchmark designed to simulate extremely challenging dose-response surfaces characterized by: (i) a highly non-linear mapping for the optimal dose $t^*(\mathbf{x})$, (ii) asymmetric penalties for deviating from the optimum, and (iii) dynamic, covariate-dependent sharpness. Covariates are sampled from a uniform distribution: $\mathbf{x} \sim \text{Unif}(-2, 2)^8$. The optimal dose $t^*(\mathbf{x})$ is generated via a two-layer random neural network. This architecture induces a complex, potentially oscillatory dependence on $\mathbf{x}$:

$$\mathbf{h}(\mathbf{x}) = \sin(\mathbf{W}_1^\top \mathbf{x} + \mathbf{b}_1), \quad t^*(\mathbf{x}) = 0.1 + 0.8 \cdot \sigma\left(\mathbf{w}_2^\top \mathbf{h}(\mathbf{x}) + b_2\right).$$

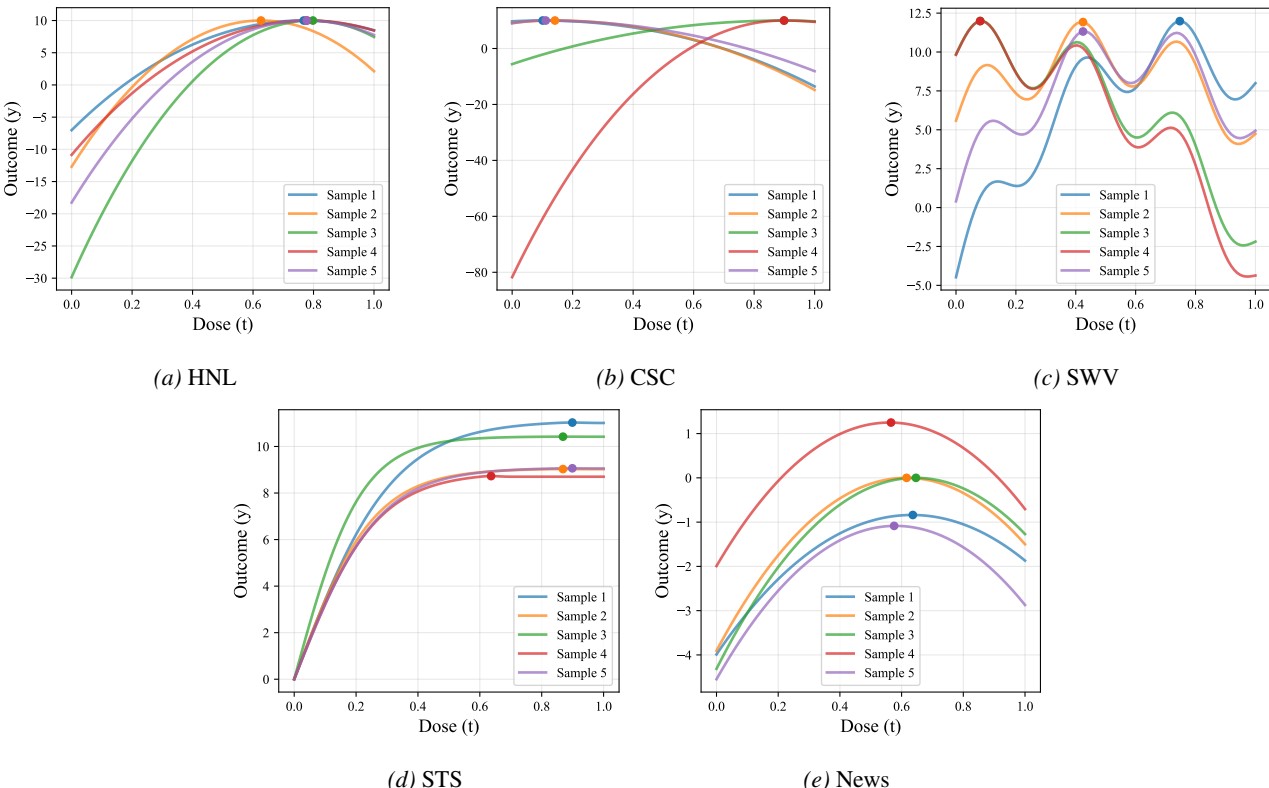

*Figure 6.* Individual dose-response curves $f(\mathbf{x}, t)$ for randomly sampled individuals across the five benchmark datasets. These curves illustrate the high degree of heterogeneity in optimal dose locations, peak sharpness, and functional forms that the algorithms should capture.

The weights $\mathbf{W}_1 \in \mathbb{R}^{8 \times 16}$ and $\mathbf{w}_2 \in \mathbb{R}^{16}$ are sampled from Unif$(-1, 1)$. The response surface is modeled using an asymmetric hyperbolic cosine function to represent different regret profiles for under- and over-dosing:

$$f(\mathbf{x}, t) = 10.0 - S(\mathbf{x}) \left( \cosh\left( \alpha(\mathbf{x}, t)\big(t - t^*(\mathbf{x})\big)\right) - 1 \right),$$

where the effective slope $\alpha(\mathbf{x}, t)$ is determined by:

$$\alpha(\mathbf{x}, t) = \begin{cases} \alpha_L(\mathbf{x}) & \text{if } t < t^*(\mathbf{x}), \\ \alpha_R(\mathbf{x}) & \text{if } t \geq t^*(\mathbf{x}). \end{cases}$$

Here, $\alpha_L(\mathbf{x}) = 0.5 + \sigma(\mathbf{x}^\top \mathbf{w}_{\text{skew}})$ and $\alpha_R(\mathbf{x}) = 2.0 - \alpha_L(\mathbf{x})$, ensuring that the sum of side-specific curvatures remains balanced while their ratio varies with $\mathbf{x}$. The projection vector $\mathbf{w}_{\text{skew}}$ is sampled from Unif$(-0.5, 0.5)^8$. The sharpness factor $S(\mathbf{x})$ controls the overall scale of the penalty:

$$S(\mathbf{x}) = 20.0 + 130.0 \cdot \sigma(\mathbf{x}^\top \mathbf{w}_{\text{sharp}}),$$

where $\mathbf{w}_{\text{sharp}} \in \mathbb{R}^8$ is another random projection vector sampled from Unif$(-0.5, 0.5)^8$. With $S(\mathbf{x})$ reaching up to 150, the peaks of the response surface can be extremely narrow, making the discovery of the optimal dose area via random exploration highly improbable.

### F.3. Simple Wavy Dataset (SWV)

The SWV dataset is an 8-dimensional synthetic benchmark that superimposes a smooth concave envelope with a high-frequency sinusoidal component over the treatment axis, resulting in dose-response curves that exhibit multiple local maxima.

Covariates are sampled uniformly as $\mathbf{x} \sim \text{Unif}(-2, 2)^8$. The core response function takes the form:

$$f(\mathbf{x}, t) = 10.0 - k(\mathbf{x})\big(\cosh\big(t - t_{\text{center}}(\mathbf{x})\big) - 1\big) + 2.0\sin(6\pi t).$$

The first term defines a symmetric concave envelope centered at $t_{\text{center}}(\mathbf{x})$, while the sine term introduces three full oscillations over the treatment interval $t \in [0, 1]$, creating multiple competing local optima. Both the envelope center $t_{\text{center}}(\mathbf{x})$ and the curvature $k(\mathbf{x})$ depend on linear projections of $\mathbf{x}$ transformed via sigmoids:

$$t_{\text{center}}(\mathbf{x}) = 0.1 + 0.7\,\sigma\big(5(\mathbf{x}^\top \mathbf{w}_{\text{center}})\big), \quad k(\mathbf{x}) = 20.0 + 40.0\,\sigma\big(\mathbf{x}^\top \mathbf{w}_{\text{sharp}}\big),$$

where $\mathbf{w}_{\text{center}}$ and $\mathbf{w}_{\text{sharp}}$ are independent random normalized vectors. A key property of SWV is that the true optimal dose $t^*(\mathbf{x})$ does not generally coincide with $t_{\text{center}}(\mathbf{x})$ because of the sinusoidal interference. Therefore, the ground-truth $t^*(\mathbf{x})$ is computed numerically via a fine-grained grid search over $t \in [0, 1]$, typically using a grid of 201 equally spaced points. This Monte Carlo-style argmax procedure provides an accurate reference for evaluating regret and prediction error, while posing a non-trivial optimization landscape for learning algorithms.

### F.4. Smooth Tanh Saturation Dataset (STS)

Following the OBD literature, which recognizes plateau-shaped dose-response patterns where efficacy saturates while additional dose escalation may increase toxicity risk (Guo & Yuan, 2023), STS models a scenario in which efficacy initially increases with dose but provides limited additional benefit beyond a saturation level. Unlike HNL and CSC, which assume a sharply peaked optimum, STS introduces a flat optimal region, allowing us to assess the general applicability of our method when strict concavity and unique-optimum assumptions are relaxed.

We sample covariates as $\mathbf{x} \sim \text{Unif}(-1, 1)^8$. Let $\sigma(a) = (1 + \exp(-a))^{-1}$. For each individual, the physical plateau point, saturation level, and growth rate are defined by

$$\bar{t}^\star(\mathbf{x}) = t_{\min} + (t_{\max} - t_{\min})\sigma(5\mathbf{x}^\top \mathbf{w}_t), \quad L(\mathbf{x}) = 8 + 4\sigma(\mathbf{x}^\top \mathbf{w}_L), \quad k(\mathbf{x}) = 4 + \sin(\mathbf{1}^\top \mathbf{x}),$$

where $t_{\min} = 0.7$, $t_{\max} = 1.0$, and $\mathbf{w}_t, \mathbf{w}_L \in \mathbb{R}^8$ are fixed random weights. The transition point is set as $t_{\text{trans}}(\mathbf{x}) = 0.85\bar{t}^\star(\mathbf{x})$. The noiseless response function is

$$f(\mathbf{x}, t) = \begin{cases} L(\mathbf{x})\tanh(k(\mathbf{x})t), & t < t_{\text{trans}}(\mathbf{x}), \\ P_5(t; \mathbf{x}), & t_{\text{trans}}(\mathbf{x}) \leq t < \bar{t}^\star(\mathbf{x}), \\ P_5(\bar{t}^\star(\mathbf{x}); \mathbf{x}), & t \geq \bar{t}^\star(\mathbf{x}), \end{cases}$$

where $P_5(t; \mathbf{x})$ is a quintic bridge connecting the $\tanh$ growth phase to the flat plateau. Let $t_0 = t_{\text{trans}}(\mathbf{x})$, $t_1 = \bar{t}^\star(\mathbf{x})$, $h = t_1 - t_0$, $u = (t - t_0)/h$, and $q(t; \mathbf{x}) = L(\mathbf{x})\tanh(k(\mathbf{x})t)$. Denote $q_0 = q(t_0; \mathbf{x})$, $q_1 = \partial_t q(t_0; \mathbf{x})$, and $q_2 = \partial_{tt} q(t_0; \mathbf{x})$. Then

$$P_5(t; \mathbf{x}) = q_0 + q_1 h u + \frac{1}{2}q_2 h^2 u^2 + \left(-\frac{3}{2}q_2 h^2 - 6q_1 h\right)u^3 + \left(\frac{3}{2}q_2 h^2 + 8q_1 h\right)u^4 + \left(-\frac{1}{2}q_2 h^2 - 3q_1 h\right)u^5.$$

This construction makes $f(\mathbf{x}, t)$ $C^2$-continuous at $t_{\text{trans}}(\mathbf{x})$ and enforces zero first- and second-order derivatives at $\bar{t}^\star(\mathbf{x})$. The ground-truth optimal dose is defined as $t^\star(\mathbf{x}) = \bar{t}^\star(\mathbf{x})$, the first point entering the zero-gradient plateau. Thus, STS provides a smooth non-strictly-concave benchmark with heterogeneous plateau locations, growth rates, and saturation levels.

### F.5. Semi-Synthetic News Dataset (News)

The News dataset is a semi-synthetic benchmark that couples real high-dimensional covariates with a controlled, known dose-response mechanism. The covariates are derived from a preprocessed news dataset (Schwab et al., 2020), resulting in a matrix $\mathbf{X}_{\text{raw}} \in \mathbb{R}^{n \times d}$ with $d = 501$ features. The raw non-negative counts are preprocessed via a log-transform followed by min–max normalization:

$$\tilde{\mathbf{X}} = \log(1 + \mathbf{X}_{\text{raw}}), \quad \mathbf{X}_{:,j} = \frac{\tilde{\mathbf{X}}_{:,j} - \min_i \tilde{\mathbf{X}}_{i,j}}{\max_i \tilde{\mathbf{X}}_{i,j} - \min_i \tilde{\mathbf{X}}_{i,j} + 10^{-9}}.$$

For each covariate vector $\mathbf{x}$, the noiseless response function $f(\mathbf{x}, t)$ is defined as:

$$f(\mathbf{x}, t) = g_1(\mathbf{x})\, c(\mathbf{x}) - g_2(\mathbf{x})\big(\cosh(t - c(\mathbf{x})) - 1\big) + g_3(\mathbf{x})(t - c(\mathbf{x})).$$

The centering term $c(\mathbf{x})$ is determined by a specific subset of features (indices 12–15), representing a covariate-dependent baseline optimal dose:

$$\bar{x}_{\text{cate}} = \frac{1}{4} \sum_{j=12}^{15} x_j, \quad c(\mathbf{x}) = 0.4 + 0.4\, \sigma\big(\alpha'(\bar{x}_{\text{cate}} - \mu_{\text{cate}})\big),$$

where $\mu_{\text{cate}}$ is the global mean of these features across the entire dataset, and $\alpha' = 5.0$ is a sensitivity scale. The coefficients $(g_1, g_2, g_3)$ are computed from disjoint feature subsets using random weights $\mathbf{w}_1, \mathbf{w}_2, \mathbf{w}_3$ normalized to unit $L_2$-norm:

$$g_1(\mathbf{x}) = 10\,(\mathbf{x}_{0:3}^\top \mathbf{w}_1),$$

$$g_2(\mathbf{x}) = 40\,\sigma\big(\cos(2\pi x_4)\sin(2\pi x_5) + \cos(2\pi x_6)\sin(2\pi x_7)\big),$$

$$g_3(\mathbf{x}) = 20\,\sigma\big(\cos(2\pi x_8)\sin(2\pi x_9) + \cos(2\pi x_{10})\sin(2\pi x_{11})\big) - 10.$$

This design yields smooth, unimodal but highly heterogeneous dose-response curves whose optimal dose $t^*(\mathbf{x})$ depends on sparse linear effects, nonlinear trigonometric interactions, and a covariate-dependent center $c(\mathbf{x})$. Because the covariates originate from real data while the response mechanism is known and fully controlled, News Dataset provides a realistic yet analytically tractable benchmark for evaluating methods that estimate individual-level dose-response functions in high dimensions.

## F.6. Regularity Parameter Estimation

In this section, we detail the estimation process of the regularity parameters $\mu$ and $L$ defined in Assumption 4.1 for the synthetic and semi-synthetic datasets used in our experiments. These parameters quantify the response sensitivity and structural properties of the dose-response functions $f(\mathbf{x}, t)$, providing a basis for our theoretical analysis. Appendix F.6 summarizes the estimated regularity parameters.

*Table 3.* Estimated regularity parameters for four datasets.

| Dataset | $\mu$ | $L$ |
|---------|-------|-----|
| HNL | 20.0 | 172.0 |
| CSC | 5.0 | 695.0 |
| SWV | 20.0[†] | 797.0 |
| News | 10.8 | 39.1 |

[†] Proxy value based on the trend component due to local fluctuations.

It is important to note that for environments containing high-frequency oscillations (e.g., SWV), the response surface exhibits complex local fluctuations. In such cases, the curvature $\mu$ is estimated based on the underlying trend component (proxied by the hyperbolic cosine term) to capture the global reward landscape, while $L$ is strictly bounded by incorporating the full impact of the oscillatory terms.

**HNL.** With the scaling factor $k(\mathbf{x}) \in [20, 120]$, the minimal curvature is determined by the lowest scaling value, yielding $\mu \approx 20.0$. The smoothness constant $L \approx 172.0$ is derived by evaluating the derivative extremes at the dose boundaries, where the hyperbolic cosine term reaches its maximum steepness.

**CSC.** The interaction between the sharpness coefficient $S(\mathbf{x}) \in [20, 150]$ and the asymmetry factors $\alpha_L, \alpha_R \in [0.5, 1.5]$ creates highly diverse response profiles. The minimal curvature $\mu \approx 5.0$ occurs at the flattest regions of the asymmetry scaling ($20 \times 0.5^2$). Conversely, the combination of high sharpness and steep slopes leads to significantly larger derivative bounds: $L \approx 695.0$.

**SWV.** In this environment, high-frequency sinusoidal perturbations are superimposed on a stable trend. While the trend suggests a baseline curvature of $\mu \approx 20.0$, the rapid oscillations contribute heavily to the higher-order derivatives. Consequently, we observe a substantial increase in the smoothness and variation parameters, with $L \approx 797.0$, reflecting the localized sensitivity of the response.

**News.** Derived from real-world covariates, this dataset exhibits more moderate variations compared to the synthetic benchmarks. Constrained by the range of the modulation coefficient $g_2(\mathbf{x})$ and the bounded feature space, the regularity parameters are estimated as $\mu \approx 10.8$ and $L \approx 39.1$.

## G. Implementation Details of Baselines

To ensure a rigorous and fair comparison, all baselines are implemented using a unified Gaussian Process (GP) surrogate model. Following the notation in our main text, given the current dataset $\mathcal{D}$, the GP provides a posterior predictive distribution for any candidate pair $(\mathbf{x}, t)$. We denote the posterior mean as $\mu_{\mathcal{D}}(\mathbf{x}, t)$ and the posterior variance as $\sigma_{\mathcal{D}}^2(\mathbf{x}, t) := k_{\mathcal{D}}((\mathbf{x}, t), (\mathbf{x}, t))$. Each sampler, at each acquisition round, aims to select a batch of units $\{\mathbf{x}_i\}_{i=1}^B$ and their corresponding optimal continuous dosages $\{\hat{t}_i\}_{i=1}^B$. For methods that involve a greedy treatment search, we identify the dosage $\hat{t}_{\mathcal{D}}^*(\mathbf{x})$ by maximizing the posterior mean as defined in Equation (6).

We categorize the baselines into three groups: (i) adaptation of active learning methods, (ii) adaptation of contextual bandit methods, and (iii) methods specifically designed for causal inference. For all methods utilizing a treatment search, we discretize the treatment space $\mathcal{T}$ into a dense uniform grid $\mathcal{T}_c$. To maintain experimental consistency, global learning settings, such as the batch size $B$, are kept identical across all baselines. Furthermore, we conducted extensive hyperparameter searches for method-specific parameters to ensure optimal performance; the corresponding search spaces and configurations are detailed in the respective tables for each baseline below.

**Dose discretization and computational trade-off.** Theoretically, a finer discretization of the dose space $\mathcal{T}_c$ allows the algorithm to more closely approximate the continuous treatment setting. However, as the grid size increases, the computational cost of acquisition score evaluation and posterior updates grows significantly. In practice, we strike a balance between approximation accuracy and computational efficiency by setting the grid size to $|\mathcal{T}_c| = 100$. Our empirical sensitivity analysis shows that this resolution is sufficient to capture the underlying dose-response curvature while maintaining a manageable overhead for real-time experimental design.

### G.1. Active Learning Baselines

The active learning category comprises methods that seek to maximize the information gain from experimental data by prioritizing areas of high model uncertainty within the joint $(\mathbf{x}, t)$ space.

#### G.1.1. MAXIMUM VARIANCE (MAXVAR)

Seo et al. (2000) established the theoretical foundation for active data selection in Gaussian Processes by prioritizing points with the highest predictive uncertainty. In a standard GP framework, the predictive variance $\sigma^2(\cdot)$ depends exclusively on the input locations rather than the observed response values, as the kernel updates are determined by the spatial distribution of the training points. Consequently, this constitutes a pure exploration strategy aimed at minimizing the overall model variance across the input manifold.

In our implementation, we extend this uncertainty-based selection to the joint $(\mathbf{x}, t)$ space to identify regions where the dose-response relationship remains least understood. For each candidate unit $\mathbf{x}_i$, we first identify the treatment $\hat{t}_{\mathcal{D}}^*(\mathbf{x}_i) = \arg\max_{t \in \mathcal{T}_c} \sigma_{\mathcal{D}}^2(\mathbf{x}_i, t)$ that maximizes the local epistemic uncertainty. Units are then ranked according to their maximum predictive variance $s_i = \sigma_{\mathcal{D}}^2(\mathbf{x}_i, \hat{t}_{\mathcal{D}}^*(\mathbf{x}_i))$. By selecting the top-$B$ units, the algorithm concentrates experimental resources on individual-dose configurations that yield the most significant reduction in the surrogate model's uncertainty, regardless of the predicted outcome.

#### G.1.2. STOCHASTIC BATCH ACQUISITION (SOFT TOP-K)

Kirsch et al. (2023) introduced stochastic batch acquisition to mitigate the "batch effect" in active learning, where greedy selection of the top-$B$ candidates often results in information redundancy due to the high correlation between neighboring points. By transforming acquisition scores into a probability distribution via a softmax function, the method introduces controlled stochasticity. This probabilistic approach naturally promotes diversity within the selected batch, ensuring that the model explores multiple regions of the input space simultaneously.

We adapt this stochastic logic to the individualized dosing task by utilizing acquisition scores $s_i$ derived from the joint $(\mathbf{x}, t)$ space. For each candidate unit $\mathbf{x}_i$, we first determine the optimal treatment $\hat{t}_{\mathcal{D}}^*(\mathbf{x}_i)$ and its corresponding acquisition score

$s_i = \sigma_{\mathcal{D}}^2(\mathbf{x}_i, \hat{t}_{\mathcal{D}}^*(\mathbf{x}_i))$ (e.g., maximum predictive variance) through an exhaustive grid search over $\mathcal{T}_c$. To prevent the batch from clustering within a single high-uncertainty region, we compute a sampling probability

$$p_i = \exp(s_i/\tau) / \sum_j \exp(s_j/\tau), \tag{55}$$

where $\tau$ denotes the temperature parameter. Rather than employing greedy ranking, $B$ units are sampled without replacement according to $p_i$. This ensures that the resulting batch $\mathcal{B}$ covers a broader spectrum of individual-dose profiles, maintaining robust exploration through probabilistic diversity. We conducted a grid search for the temperature parameter $\tau$ to balance greedy uncertainty maximization and batch diversity, as detailed in Table 4.

*Table 4.* Hyperparameter search space for the Soft Top-K sampler.

| Hyperparameter | Range (All Datasets) |
|---|---|
| Temperature $\tau$ | 0.01, 0.05, 0.1, 0.5, 1.0 |

## G.2. Contextual Bandit Baselines

The following baselines are adapted from the contextual bandit literature. In our framework, unit features $\mathbf{x}$ serve as the context and dosage $t$ as the continuous action. Each baseline identifies an optimal dosage $\hat{t}_i^*$ by maximizing its respective acquisition function conditioned on $\mathcal{D}$.

### G.2.1. BATCHED THOMPSON SAMPLING (TS)

Kalkanli & Ozgur (2021) investigated the theoretical regret bounds of Thompson Sampling (TS) in batched settings. The core principle of TS is probability matching: at each iteration, a function $\tilde{f}$ is sampled from the posterior distribution, and the action that maximizes this sample is selected. This approach naturally balances exploration and exploitation by selecting actions according to the probability that they are optimal under the current posterior beliefs.

To adapt this to individualized dosing, we leverage the reparameterization trick to generate stochastic responses:

$$\tilde{f}(\mathbf{x}, t) = \mu_{\mathcal{D}}(\mathbf{x}, t) + \sigma_{\mathcal{D}}(\mathbf{x}, t) \cdot \epsilon, \tag{56}$$

where $\epsilon \sim \mathcal{N}(0, 1)$. For each candidate $\mathbf{x}_i$ in the sampled pool $\mathcal{X}_c$, we perform an exhaustive search over the grid $\mathcal{T}_c$ to find the optimal dosage for that specific individual: $\hat{t}_i^* = \arg\max_{t \in \mathcal{T}_c} \tilde{f}(\mathbf{x}_i, t)$. Units are then ranked by their sampled scores $s_i = \tilde{f}(\mathbf{x}_i, \hat{t}_i^*)$, and the top-$B$ units are selected to form the experimental batch.

### G.2.2. BATCHED GP-UCB (GP-UCB)

Li & Scarlett (2022) explored GP-UCB performance in settings with a limited number of interaction batches. The algorithm balances exploration and exploitation by maximizing the Upper Confidence Bound (UCB), where the acquisition score is defined as the predictive mean $\mu$ plus a weighted standard deviation $\beta^{1/2}\sigma$. This weighted term, $\beta^{1/2}\sigma$, serves as a theoretical parameter scaling the information gain based on the model's epistemic uncertainty.

We extend the UCB score to the joint $(\mathbf{x}, t)$ space via the acquisition function

$$\alpha_{\mathrm{UCB}}(\mathbf{x}, t) = \mu_{\mathcal{D}}(\mathbf{x}, t) + \beta^{1/2} \cdot \sigma_{\mathcal{D}}(\mathbf{x}, t). \tag{57}$$

For each candidate $\mathbf{x}_i$, we determine the best possible confidence bound achievable by searching the grid $\mathcal{T}_c$: $\hat{t}_i^* = \arg\max_{t \in \mathcal{T}_c} \alpha_{\mathrm{UCB}}(\mathbf{x}_i, t)$. The final score for the unit is assigned as $s_i = \alpha_{\mathrm{UCB}}(\mathbf{x}_i, \hat{t}_i^*)$. By selecting units with the highest $s_i$, the sampler identifies individuals who exhibit the highest potential response under their respective optimal predicted dosing. The exploration-exploitation trade-off in GP-UCB is primarily governed by the parameter $\beta$. We performed a hyperparameter search over its values to identify the optimal balance for each dataset, as summarized in Table 5.

### G.2.3. CONTEXTUAL EXPECTED IMPROVEMENT (EI)

Gupta et al. (2022) derived the Expected Improvement (EI) metric specifically for contextual bandits. EI measures the expected progress over the current best-observed value $y_{\max}$. In contextual settings, the acquisition must be conditioned on

*Table 5.* Hyperparameter search space for the GP-UCB sampler.

| Hyperparameter | Range (All Datasets) |
|---|---|
| Exploration weight $\beta$ | 0.25, 1.0, 4.0, 9.0, 16.0 |

the environmental information (context) to select the optimal action, effectively narrowing the search to regions likely to yield an improvement over the current state-of-the-art.

We frame the unit features $\mathbf{x}$ as the decision context and utilize the closed-form expression for GP-EI:

$$\text{EI}(\mathbf{x}, t) = \Delta \Phi \left( \frac{\Delta}{\sigma_{\mathcal{D}}(\mathbf{x}, t)} \right) + \sigma_{\mathcal{D}}(\mathbf{x}, t) \cdot \phi \left( \frac{\Delta}{\sigma_{\mathcal{D}}(\mathbf{x}, t)} \right). \tag{58}$$

Here, $\Delta(\mathbf{x}, t) := \mu_{\mathcal{D}}(\mathbf{x}, t) - (y_{\max} + \xi)$ represents the expected improvement margin, in which $\xi \geq 0$ is a hyperparameter that controls the trade-off between exploration and exploitation. Additionally, $\Phi(\cdot)$ and $\phi(\cdot)$ denote the cumulative distribution function and the probability density function of the standard normal distribution, respectively.

To handle the individualized dosing setting, we implement a two-step selection strategy: (i) Treatment Optimization: For each candidate unit $\mathbf{x}_i$ in the candidate pool $\mathcal{X}_c$, we first perform a grid search over $\mathcal{T}_c$ to identify the treatment $\hat{t}_i^* = \arg\max_{t \in \mathcal{T}_c} \text{EI}(\mathbf{x}_i, t)$ that yields the maximum expected improvement for that specific individual. (ii) Unit Ranking: The units are then ranked based on these optimized acquisition scores $s_i = \text{EI}(\mathbf{x}_i, \hat{t}_i^*)$. By selecting the top-$B$ units with the highest peak EI values, the algorithm prioritizes individuals who are most likely to surpass current performance benchmarks when assigned their respective optimized dosages. The exploration parameter $\xi$ balances local improvement and global uncertainty in the Contextual EI sampler. Its search space is detailed in Table 6.

*Table 6.* Hyperparameter search space for the EI sampler.

| Hyperparameter | Range (All Datasets) |
|---|---|
| Exploration parameter $\xi$ | 0.01, 0.05, 0.1, 0.2 |

### G.2.4. CONTINUOUS ACTION TREE WITH SMOOTHING (CATS)

Majzoubi et al. (2020) proposed the CATS framework to handle continuous action spaces by adaptively partitioning the action range into a hierarchical tree structure. A key intuition of CATS is that information should be shared locally within the action space to mitigate data sparsity and surrogate model uncertainty. By smoothing rewards across adjacent nodes in the tree, the algorithm ensures a more robust policy learning process in continuous domains.

In our implementation, we adapt the CATS principle by substituting the dynamic tree structure with the dense treatment grid $\mathcal{T}_c$. The selection process follows a two-stage adaptive logic: (i) Unit Selection: To maintain diversity and follow the adaptive spirit of the baseline, a batch of $B$ units $\{\mathbf{x}_i\}_{i=1}^B$ is first identified from the candidate pool (e.g., via random sampling). (ii) Localized Smoothing: For each selected unit, we identify the greedy dosage $\hat{t}_{\mathcal{D}}^*(\mathbf{x})$ by maximizing $\mu_{\mathcal{D}}(\mathbf{x}, t)$ and then sample the final treatment $t$ from a localized interval:

$$t \sim \text{Uniform}(\max(0, \hat{t}_{\mathcal{D}}^*(\mathbf{x}) - h), \min(1, \hat{t}_{\mathcal{D}}^*(\mathbf{x}) + h)), \tag{59}$$

where $h$ is the smoothing bandwidth. This approach ensures the assignment is robust to local surrogate fluctuations while exploring variations in the optimal neighborhood. We conducted a hyperparameter search for $h$ to ensure optimal robust policy learning, as summarized in Table 7.

### G.2.5. POLICY LEARNING WITH ADAPTIVE SAMPLING (PLAS)

PLAS (Kato et al., 2024) is a recent contextual best-arm identification baseline for adaptive experimental design under exogenously arriving contexts. To adapt it to our pool-based one-shot intervention setting, we preserve this exogenous-context mechanism by first sampling a batch of $B$ distinct units uniformly from the available pool $X_{\text{avail}}$. Since PLAS cannot directly handle continuous dosages, we adapt it to our setting via dosage discretization. For each selected unit $\mathbf{x}_i$, we

*Table 7.* Hyperparameter search space for the CATS sampler.

| Hyperparameter | Range (All Datasets) |
|---|---|
| Smoothing Bandwidth $h$ | 0.01, 0.02, 0.05, 0.1, 0.2 |

discretize the continuous dosage space into $\mathcal{T}_c = \{t_1, \ldots, t_K\}$ and evaluate the GP posterior variance $v_{ik} = \sigma_D^2(\mathbf{x}_i, t_k)$, $k = 1, \ldots, K$. Following the PLAS allocation rule, these variances are converted into a treatment-assignment distribution

$$p_{ik} = \frac{v_{ik}}{\sum_{\ell=1}^{K} v_{i\ell}}, \tag{60}$$

with a small numerical floor applied to $v_{ik}$ for stability. The assigned dosage is then sampled as

$$t_i \sim \text{Categorical}(p_{i1}, \ldots, p_{iK}), \qquad k = 1, \cdots, K, \tag{61}$$

over the discretized treatment grid. Thus, unlike acquisition functions that jointly optimize over $(\mathbf{x}, t)$, PLAS keeps unit selection random and concentrates adaptivity in the treatment-allocation distribution, assigning higher probability to dosages where the current GP surrogate remains more uncertain. In our experiments, PLAS uses the same GP surrogate and treatment grid as the other baselines and introduces no additional tuned hyperparameter beyond the grid resolution.

### G.3. Baselines Designed for Causal Inference

This section introduces methods explicitly formulated to minimize treatment estimation errors or to optimize information theoretic objectives specific to treatment effect modeling.

#### G.3.1. ACTIVE BAYESIAN CAUSAL INFERENCE WITH COHN CRITERIA (ABC3)

The ABC3 proposed by Cha & Lee (2025) adopts the classic Cohn criteria (Cohn et al., 1996) to select samples that maximize the reduction in the model's global predictive uncertainty. Theoretically, the method demonstrates that minimizing the estimation error of the Conditional Average Treatment Effect (CATE) is equivalent to minimizing the Integrated Posterior Variance (IPV) of the estimator. Leveraging the analytical properties of Gaussian Processes, the method derives a closed-form expression for the expected variance reduction, which serves as the acquisition function for selecting the next optimal intervention.

To adapt the original binary treatment framework to a continuous dosage space $\mathcal{T}$, we implement a grid-based sampling mechanism. Following the global uncertainty reduction objective, the optimal pair $(\mathbf{x}^*, \hat{t}^*)$ is selected from the query set $\mathcal{Q}$ as follows:

$$(\mathbf{x}^*, \hat{t}^*) = \arg \max_{(\mathbf{x}, t) \in \mathcal{Q}} s(\mathbf{x}, t), \qquad s(\mathbf{x}, t) := \frac{\int_{\mathcal{X} \times \mathcal{T}} [k_\mathcal{D}((\mathbf{x}, t), (\mathbf{x}', t'))]^2 d\mathbb{P}(\mathbf{x}', t')}{\sigma_\mathcal{D}^2(\mathbf{x}, t) + \sigma_\epsilon^2}, \tag{62}$$

where $\sigma_\epsilon^2$ is the variance of the additive observation noise defined in Equation (1). We approximate the integral over the domain by drawing a representative reference set $\mathcal{Z}$ from the full candidate pool $X_{pool}$ and partitioning the treatment space into the uniform grid $\mathcal{T}_c$. This discretization transforms the continuous optimization into a computationally feasible search while maintaining numerical stability. To enable efficient batch selection, we employ a greedy strategy where scores are computed for all candidate-dosage pairs in the search space $\mathcal{Q}$, and the top-$B$ pairs $\{(\mathbf{x}_b, t_b)\}_{b=1}^B$ with the highest variance reduction scores $s$ are selected to form the experimental batch.

#### G.3.2. POLICY GRADIENT FOR EXPECTED INFORMATION GAIN (PG)

**Method Overview.** The Policy Gradient for Expected Information Gain algorithm proposed by Razzak et al. (2024) aims to learn a joint policy $\pi_\phi(X, T)$ in each acquisition round to maximize the expected information gain (EIG) of a full experimental batch. In the inner optimization loop, the algorithm iteratively samples a batch $\{(\mathbf{x}_b, t_b)\}_{b=1}^B$ from the current policy, computes the reward $R$, and performs gradient ascent updates.

The joint policy is decomposed into $\pi_\phi(X, T) = \pi_{\phi_\mathbf{x}}(X) \pi_{\phi_t}(T \mid X)$. For discrete instance selection $\pi_{\phi_\mathbf{x}}$, a sequential masked categorical distribution is employed over the finite candidate pool to satisfy the one-shot constraint. The parameters

$\phi_{\mathbf{x}}$ are optimized using the REINFORCE algorithm with an exponential moving average (EMA) baseline $b$, where the advantage is defined as $A = R - b$. For continuous treatments, a reparameterizable Logit-Normal policy is deployed: $t_b = \mathrm{sigmoid}(\mu_{\phi_t}(\mathbf{x}_b) + \exp(\rho_{\phi_t})\epsilon_b)$, where $\epsilon_b \sim \mathcal{N}(0, 1)$. Here, $\mu_{\phi_t}$ and $\rho_{\phi_t}$ represent the learnable mean and a shared log-standard deviation, respectively. This formulation constructs a differentiable path from the reward $R$ to the parameters $\phi_t$, enabling efficient optimization via pathwise gradients.

**Adaptation and Implementation Details.** In our baseline implementation, the original deep learning surrogate is replaced with a Gaussian Process (GP). Leveraging the analytical properties of GPs, the reward is computed using closed-form information gain, eliminating the need for the original Monte Carlo approximation:

$$R = \frac{1}{2} \log \det(\mathbf{I} + \mathbf{K}/\sigma^2)$$

where $\mathbf{K}$ is the posterior predictive covariance matrix. In practice, this is computed via Cholesky decomposition with jitter for numerical stability (equivalent to maximizing $\frac{1}{2}\log\det(\mathbf{K} + \sigma^2\mathbf{I})$). The final joint optimization objective combines the policy gradient term for discrete variables and the reparameterization term for continuous variables:

$$\mathcal{L} = -A \sum_{b=1}^{B} \log \pi_{\phi_{\mathbf{x}}}(\mathbf{x}_b \mid \mathbf{x}_{<b}) \; - \; R,$$

where $\mathbf{x}_{<b} = \{\mathbf{x}_1, \dots, \mathbf{x}_{b-1}\}$ denotes the set of units selected prior to the $b$-th step within the same batch (with $\mathbf{x}_{<1} = \emptyset$). Parameters $\phi_x$ and $\phi_t$ are updated via the Adam optimizer. We performed a hyperparameter search for the respective learning rates $l_x$ and $l_t$ to ensure optimal baseline performance, as listed in Table 8.

*Table 8.* Hyperparameter search space for the PG sampler.

| Hyperparameter | Range (All Datasets) |
|---|:---:|
| Optimizer | Adam |
| Inner optimization epochs | 50 |
| Learning rate $\eta_x$ (for $\phi_x$) | 0.1, 0.01, 0.001 |
| Learning rate $\eta_t$ (for $\phi_t$) | 0.1, 0.01, 0.001 |

# H. Additional Results

## H.1. Additional Results on Baselines Compare

The following Tables 9 to 11 present a more detailed evaluation of policy suboptimality on the three main datasets, complementing the selected query-budget results reported in Table 1.

## H.2. Additional Results on Initialization Ratio Experiment

In this appendix, we provide supplementary experimental results and analyzes regarding initialization strategies. First, for high-dimensional complex datasets such as News, the performance tends to exhibit a characteristic U-shaped trend across all methods (Figure 7d). This pattern indicates that a moderate warm start is often required to prevent initial model instability; however, GVALID consistently maintains the most robust performance across nearly all initialization ratios.

Furthermore, we evaluate the impact of different sampling techniques by comparing Latin Hypercube Sampling (LHS) with Random Sampling (RAND). The results demonstrate that LHS provides a distinct advantage over RAND, particularly when the initialization ratio is low (Figure 8). By ensuring a more uniform coverage of the covariate space, LHS effectively mitigates initial model uncertainty, leading to superior policy suboptimality ($\mathcal{R}$) in the early stages of the learning process.

## H.3. Sensitivity to the Finite-Difference Step Size

In this section, we study the sensitivity of GVALID to the finite-difference step size $\delta$, where $\delta$ is the perturbation radius used to construct the symmetric finite-difference gradient proxy defined in Equation (11). We evaluate $\delta \in$

*Table 9.* Policy suboptimality performance ($\mathcal{R}$) of various methods at different number of queries($N$) on HNL dataset.

| METHOD | N=100 | N=150 | N=200 | N=250 | N=300 | N=350 | N=400 |
|---|---|---|---|---|---|---|---|
| TS | $0.90 \pm 0.43$ | $0.56 \pm 0.21$ | $0.47 \pm 0.20$ | $0.39 \pm 0.12$ | $0.36 \pm 0.13$ | $0.31 \pm 0.11$ | $0.29 \pm 0.11$ |
| GP-UCB | $0.98 \pm 0.35$ | $0.78 \pm 0.38$ | $0.62 \pm 0.27$ | $0.58 \pm 0.27$ | $0.51 \pm 0.21$ | $0.46 \pm 0.16$ | $0.42 \pm 0.16$ |
| EI | $1.33 \pm 0.61$ | $0.90 \pm 0.35$ | $0.82 \pm 0.39$ | $0.68 \pm 0.23$ | $0.60 \pm 0.18$ | $0.55 \pm 0.20$ | $0.53 \pm 0.19$ |
| CATS | $0.95 \pm 0.34$ | $0.71 \pm 0.27$ | $0.55 \pm 0.20$ | $0.43 \pm 0.11$ | $0.36 \pm 0.09$ | $0.31 \pm 0.08$ | $0.29 \pm 0.08$ |
| MAXVAR | $0.62 \pm 0.23$ | $0.41 \pm 0.17$ | $0.33 \pm 0.12$ | $0.25 \pm 0.09$ | $0.20 \pm 0.07$ | $0.18 \pm 0.06$ | $0.17 \pm 0.05$ |
| ABC3 | $0.61 \pm 0.26$ | $0.43 \pm 0.15$ | $0.31 \pm 0.11$ | $0.24 \pm 0.09$ | $0.21 \pm 0.07$ | $0.17 \pm 0.05$ | $0.15 \pm 0.05$ |
| PG | $0.69 \pm 0.23$ | $0.54 \pm 0.18$ | $0.43 \pm 0.14$ | $0.40 \pm 0.09$ | $0.34 \pm 0.07$ | $0.33 \pm 0.07$ | $0.30 \pm 0.07$ |
| SOFTTOPK | $0.56 \pm 0.25$ | $0.43 \pm 0.17$ | $0.32 \pm 0.11$ | $0.25 \pm 0.09$ | $0.21 \pm 0.06$ | $0.18 \pm 0.06$ | $0.17 \pm 0.05$ |
| PLAS | $0.67 \pm 0.28$ | $0.49 \pm 0.17$ | $0.42 \pm 0.15$ | $0.41 \pm 0.12$ | $0.37 \pm 0.11$ | $0.33 \pm 0.09$ | $0.32 \pm 0.08$ |
| RAND | $0.73 \pm 0.32$ | $0.54 \pm 0.16$ | $0.41 \pm 0.12$ | $0.40 \pm 0.11$ | $0.36 \pm 0.09$ | $0.33 \pm 0.10$ | $0.32 \pm 0.09$ |
| GVALID | $\mathbf{0.57 \pm 0.33}$ | $\mathbf{0.40 \pm 0.24}$ | $\mathbf{0.23 \pm 0.10}$ | $\mathbf{0.18 \pm 0.09}$ | $\mathbf{0.15 \pm 0.06}$ | $\mathbf{0.12 \pm 0.04}$ | $\mathbf{0.11 \pm 0.04}$ |

*Table 10.* Policy suboptimality performance ($\mathcal{R}$) of various methods at different number of queries($N$) on CSC dataset.

| METHOD | N=100 | N=150 | N=200 | N=250 | N=300 | N=350 | N=400 |
|---|---|---|---|---|---|---|---|
| TS | $1.46 \pm 0.82$ | $1.15 \pm 0.52$ | $0.98 \pm 0.41$ | $0.87 \pm 0.42$ | $0.79 \pm 0.36$ | $0.72 \pm 0.34$ | $0.63 \pm 0.30$ |
| GP-UCB | $2.15 \pm 1.68$ | $1.78 \pm 1.73$ | $1.82 \pm 2.35$ | $1.34 \pm 1.12$ | $0.99 \pm 0.45$ | $0.91 \pm 0.43$ | $0.87 \pm 0.39$ |
| EI | $2.92 \pm 3.29$ | $2.45 \pm 2.97$ | $2.97 \pm 3.98$ | $2.38 \pm 3.48$ | $2.10 \pm 3.27$ | $2.08 \pm 3.32$ | $1.73 \pm 3.00$ |
| CATS | $1.62 \pm 0.77$ | $1.26 \pm 0.62$ | $1.04 \pm 0.45$ | $0.95 \pm 0.65$ | $0.77 \pm 0.47$ | $0.66 \pm 0.28$ | $0.64 \pm 0.29$ |
| MAXVAR | $\mathbf{1.25 \pm 0.71}$ | $0.87 \pm 0.41$ | $0.76 \pm 0.39$ | $0.65 \pm 0.26$ | $0.56 \pm 0.22$ | $0.50 \pm 0.18$ | $0.46 \pm 0.17$ |
| ABC3 | $1.26 \pm 0.73$ | $0.86 \pm 0.40$ | $0.78 \pm 0.32$ | $0.67 \pm 0.25$ | $0.59 \pm 0.19$ | $0.55 \pm 0.18$ | $0.50 \pm 0.18$ |
| PG | $1.76 \pm 1.65$ | $0.91 \pm 0.38$ | $0.74 \pm 0.31$ | $0.63 \pm 0.22$ | $0.56 \pm 0.19$ | $0.51 \pm 0.16$ | $0.46 \pm 0.16$ |
| SOFTTOPK | $1.78 \pm 2.51$ | $1.57 \pm 2.62$ | $1.35 \pm 2.63$ | $1.22 \pm 2.63$ | $1.15 \pm 2.53$ | $1.08 \pm 2.42$ | $1.02 \pm 2.38$ |
| PLAS | $1.60 \pm 1.25$ | $0.94 \pm 0.37$ | $0.74 \pm 0.29$ | $0.69 \pm 0.28$ | $0.58 \pm 0.24$ | $0.53 \pm 0.19$ | $0.50 \pm 0.21$ |
| RAND | $1.55 \pm 1.38$ | $1.00 \pm 0.62$ | $0.79 \pm 0.33$ | $0.71 \pm 0.31$ | $0.60 \pm 0.23$ | $0.56 \pm 0.22$ | $0.53 \pm 0.25$ |
| GVALID | $1.61 \pm 1.15$ | $\mathbf{0.84 \pm 0.53}$ | $\mathbf{0.68 \pm 0.49}$ | $\mathbf{0.50 \pm 0.32}$ | $\mathbf{0.48 \pm 0.31}$ | $\mathbf{0.40 \pm 0.26}$ | $\mathbf{0.35 \pm 0.20}$ |

$\{0.005, 0.01, 0.02, 0.05, 0.1\}$ on the CSC dataset and additionally compare with the exact GP derivative posterior. Although exact GP derivatives are analytically available, they require derivative-related covariance computations in the batched acquisition loop, whereas finite differences only use standard GP posterior evaluations at perturbed dose locations. As shown in Figure 9, moderate choices of $\delta$ achieve more stable and competitive policy suboptimality, while an overly small step size leads to slower improvement. Notably, the exact derivative variant does not outperform finite differences and even exhibits slower improvement, suggesting that analytical gradients are not necessarily preferable in low-data active learning regimes.

This behavior reflects a practical bias variance trade-off in gradient approximation. A very small $\delta$ is closer to the infinitesimal derivative in principle, but it may amplify posterior estimation noise in early active learning rounds when the GP surrogate is still uncertain. Conversely, a large $\delta$ may introduce truncation bias by averaging responses over a wider neighborhood around the estimated optimal dose. The inferior performance of exact GP derivatives further suggests that analytical gradients can be sensitive to noisy posterior estimates in low-data regimes, whereas finite differences provide a smoothed local gradient signal for more robust gradient-variance-based acquisition. Overall, GVALID is stable within a moderate range of $\delta$, and we use a fixed moderate value in the main experiments.

*Table 11.* Policy suboptimality performance ($\mathcal{R}$) of various methods at different number of queries($N$) on SWV dataset.

| METHOD | N=100 | N=150 | N=200 | N=250 | N=300 | N=350 | N=400 |
|---|---|---|---|---|---|---|---|
| TS | $1.00 \pm 0.27$ | $0.79 \pm 0.20$ | $0.63 \pm 0.18$ | $0.56 \pm 0.16$ | $0.49 \pm 0.14$ | $0.43 \pm 0.11$ | $0.38 \pm 0.08$ |
| GP-UCB | $1.36 \pm 0.59$ | $0.97 \pm 0.33$ | $0.89 \pm 0.22$ | $0.74 \pm 0.21$ | $0.66 \pm 0.17$ | $0.57 \pm 0.15$ | $0.56 \pm 0.22$ |
| EI | $1.47 \pm 0.57$ | $1.13 \pm 0.39$ | $0.89 \pm 0.19$ | $0.76 \pm 0.17$ | $0.71 \pm 0.16$ | $0.62 \pm 0.13$ | $0.56 \pm 0.12$ |
| CATS | $1.14 \pm 0.54$ | $0.84 \pm 0.29$ | $0.67 \pm 0.25$ | $0.56 \pm 0.20$ | $0.45 \pm 0.16$ | $0.38 \pm 0.14$ | $0.34 \pm 0.12$ |
| MAXVAR | $1.25 \pm 0.47$ | $0.96 \pm 0.42$ | $0.71 \pm 0.33$ | $0.52 \pm 0.23$ | $0.45 \pm 0.21$ | $0.35 \pm 0.11$ | $0.31 \pm 0.10$ |
| ABC3 | $1.09 \pm 0.50$ | $0.79 \pm 0.28$ | $0.53 \pm 0.13$ | $0.43 \pm 0.14$ | $0.38 \pm 0.10$ | $0.36 \pm 0.10$ | $0.32 \pm 0.09$ |
| PG | $0.91 \pm 0.26$ | $0.67 \pm 0.16$ | $0.58 \pm 0.15$ | $0.50 \pm 0.11$ | $0.45 \pm 0.12$ | $0.38 \pm 0.11$ | $0.36 \pm 0.10$ |
| SOFTTOPK | $1.41 \pm 0.59$ | $0.84 \pm 0.37$ | $0.64 \pm 0.31$ | $0.51 \pm 0.20$ | $0.42 \pm 0.16$ | $0.35 \pm 0.10$ | $0.31 \pm 0.09$ |
| PLAS | $\mathbf{0.77 \pm 0.05}$ | $\mathbf{0.60 \pm 0.07}$ | $\mathbf{0.44 \pm 0.08}$ | $0.39 \pm 0.15$ | $0.33 \pm 0.10$ | $0.34 \pm 0.11$ | $0.32 \pm 0.15$ |
| RAND | $0.99 \pm 0.38$ | $0.71 \pm 0.23$ | $0.51 \pm 0.11$ | $0.43 \pm 0.09$ | $0.41 \pm 0.09$ | $0.37 \pm 0.08$ | $0.33 \pm 0.09$ |
| GVALID | $0.91 \pm 0.44$ | $0.65 \pm 0.19$ | $0.49 \pm 0.16$ | $\mathbf{0.37 \pm 0.12}$ | $\mathbf{0.33 \pm 0.10}$ | $\mathbf{0.29 \pm 0.10}$ | $\mathbf{0.25 \pm 0.08}$ |

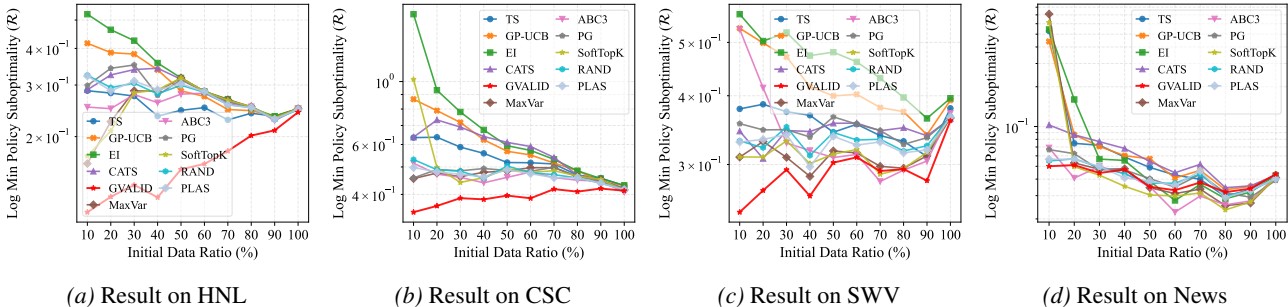

*(a)* Result on HNL     *(b)* Result on CSC     *(c)* Result on SWV     *(d)* Result on News

*Figure 7.* LHS initialization ratio impact on policy suboptimality ($\mathcal{R}$).

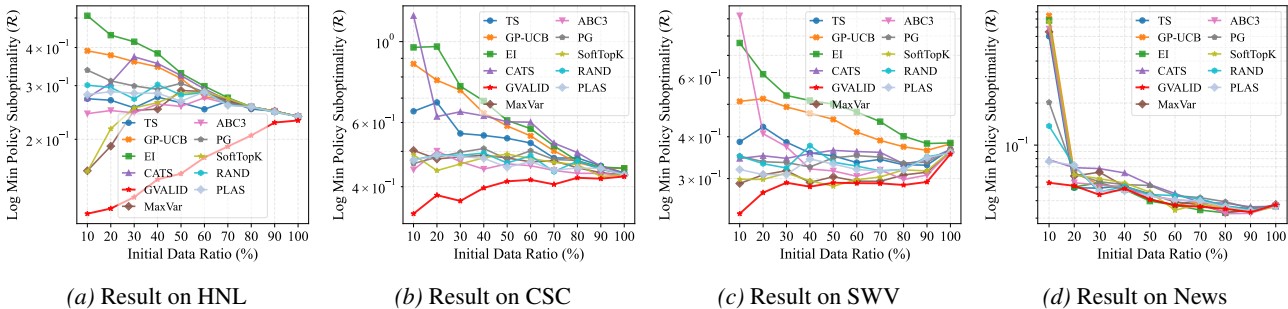

*(a)* Result on HNL     *(b)* Result on CSC     *(c)* Result on SWV     *(d)* Result on News

*Figure 8.* Rand initialization ratio impact on policy suboptimality ($\mathcal{R}$).

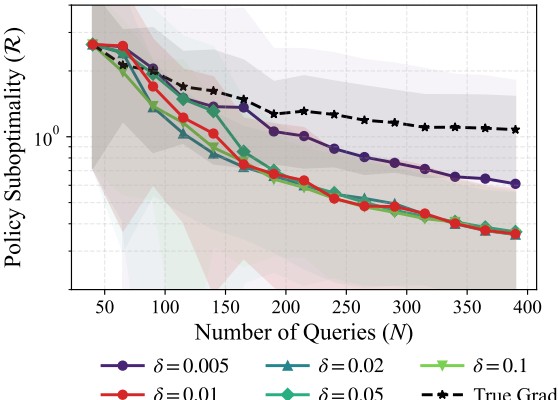

*Figure 9.* Sensitivity analysis of the finite-difference step size $\delta$ on the CSC dataset.

