# OpenReview forum: "Active Policy Optimization for Individualized Dosing via Gradient Variance Minimization"
_ICML.cc/2026/Conference — ICML 2026 regular_

### Official Review · Reviewer_yJDK · 2026-03-10

**Soundness:** 3
**Presentation:** 2
**Significance:** 2
**Originality:** 3
**Overall Recommendation:** 4
**Confidence:** 4

**Summary:**

This paper considers policy learning for individualised dosing in scenarios of concave unimodal dose-response relationships to identify the dose with largest response (optimal dose). It considers the gradient of an intervention’s dose-response curve as a surrogate for a dose’s proximity to the optimal dose. The paper proposes GVALID, a methodology which aims to reduce the uncertainty around the gradient posterior estimator instead of standard global predictive uncertainty.

**Compliance With Llm Reviewing Policy:**

Affirmed.

**Final Justification:**

I have increased my rating to Weak accept (4) as authors have sufficiently addressed limitations I identified within my initial review.

**Key Questions For Authors:**

(1)	For section "Finite-difference construction" (Lines 235-241), it is unclear to me whether individuals in $\mathbf{z}_j^{+}$ are $\mathbf{z}_j^{-}$ represent the same individual evaluated under different treatment doses (i.e., within a potential outcomes framework) or two distinct but otherwise identical individuals drawn from the pool. If they are the same individual, authors should provide more details on how potential outcomes are estimated in this paper. If they are two identical individuals, I ask how likely it is to recruit individuals with identical covariates, particularly within clinical trials where recruitment is already challenging in general.

(2)	Whilst the discussion of the SWV dataset experiment briefly discusses GVALID performance in light of assumption violations, I would be interested to see GVALID performance in identifying an optimal dose for a plateauing dose-response relationship.

(3)	Authors should more clearly acknowledge that GVALID's theoretical guarantees are restricted to unimodal dose-response relationships alone. They should further acknowledge other common dose-response relationships for OBDs.

(4)	Authors should consider evaluating the performance of GVALID in terms of global predictive uncertainty to support practitioners in evaluating the trade-off between GVALID and other approaches in terms of Bayes regret and global estimation.

**Limitations:**

Authors do not provide limitations to their work. I have provided some throughout this review which authors may wish to take on board.

**Strengths And Weaknesses:**

Strengths:

(1)	The use of a gradient uncertainty in terms of gradient posterior variance appears to be a novel contribution to active learning literature. I am satisfied that the paper is original.

(2)	 The manuscript presents an appropriate number of experiments for its initial submission across three datasets and 9 alternative methods. GVALID demonstrates superior performance for the chosen metric of the Bayes regret.

(3)	The paper is presented well and is well written. Overall sufficient related work sections are provided in both the main manuscript and Appendix.

(4)	Individualised treatment dosing is gaining increased scientific interest, particularly in the field of precision medicine. I do think this paper addresses a relevant challenge, though theoretical assumptions required to guarantee performance constrain the actual application of this methodology to specific use cases.

Weaknesses:

(1)	GVALID is motivated predominantly by dose-optimisation clinical trials to determine the optimal biological dose (OBD) (Lines 46-53, right column). Whilst authors claim dose-response relationships are often unimodal, both cited papers Zang, Lee et al. 2014 [1] and Zhang, Sargent et al. 2006 [2] recognise other dose-response relationships too, including those which are monotonically increasing/decreasing and plateauing. With theoretical guarantees only suitable for concave unimodal relationships, GVALID is only suitable for dose-decision making when a unimodal relationship is confidently supposed a priori. This may be rarely the case in early-phase clinical trials where optimisation strategies are typically explored. Authors do not sufficiently address potential limitations to their work in the manuscript which limits the applicability of the paper and impact of the research.


(2)	GVALID demonstrates superior performance to other methods when the optimal dose is the dose with the best response. However, dose selection may depend upon other outcomes e.g. toxicity in clinical trials for the identification of the OBD. Experiments demonstrating the performance of GVALID for estimation-orientated tasks are not presented. Evaluating GVALIDs performance for estimating the whole dose-response relationship remains essential, particularly in instances where GVALID is incorrectly employed for a treatment assumed to have unimodal dose-response relationship which in fact exhibits plateauing or monotonic behaviour.


(3)	Whilst the paper is positioned well in active policy optimisation literature, it could better integrated within dose-optimisation literature. Some referenced literature is misused, for example Wages and Tait 2015 [3] presents no evidence of employing non-parametric Bayesian modelling (Line 152, right column). This paper does not provide sufficient evidence to demonstrate that typical dose–response relationships exhibit adequate smoothness and differentiability across dose levels. This weakens the strength and potential significance of the paper.

(4)	Whilst response is considered a continuous outcome in this paper, it is often defined as a binary endpoint in clinical trials. As such, the probability of response may represent a more appropriate outcome measure. The paper does not stipulate whether data types other than continuous response can be used with GVALID.


Minor comments:

•	The colours used to identify each method in Figures 2 and 4 are inconsistent.

•	Please update referenced literature to contemporary discussion on treatment-response paradigms. Zang, Lee et al. 2014 [1] and Zhang, Sargent et al. 2006 [2] are over a decade old. Are there more recent works you could cite to motivate your argument in Lines 51-52, right column?

•	The function f(x,t) is only referred to as the conditional average treatment effect in the Appendix. It should also be designated as the conditional average treatment effect in the main manuscript too.

References:


[1] Zang, Y., et al. (2014). "Adaptive designs for identifying optimal biological dose for molecularly targeted agents." Clinical Trials 11(3): 319–327.

[2] Zhang, W., et al. (2006). "An adaptive dose‐finding design incorporating both toxicity and efficacy." Statistics in medicine 25(14): 2365–2383.

[3] Wages, N. A. and C. Tait (2015). "Seamless phase I/II adaptive design for oncology trials of molecularly targeted agents." Journal of Biopharmaceutical Statistics 25(5): 903–920.

---

> ### Author Rebuttal · Authors · 2026-03-30
>
> **Unimodal assumption:**
> We appreciate the feedback regarding our assumptions.
> We initially tailored our theory and method to unimodal curves due to their prevalence, though our original SWV experiments already evaluate performance beyond this assumption.
> Acknowledging that plateau-shape are also clinically common, we added experiments evaluating them.
> Additionally, our theory naturally extends to multi-peak and plateau settings (see response to Reviewer CXdr).
>
> We use Smooth Tanh Saturation (STS) model to replicate OBD-style responses:$$f(x,t)=L\tanh(kt) \mathbb{1}_ {t<t_ {\text{trans}}}+P_5(t;x) \mathbb{1}_ {t_{\text{trans}}\le t<t^{\star} } + f(x,t^{\star} )\mathbb{1}_ {t\ge t^{\star}}$$
> Here, $P_5$ quintically bridges the Tanh to a plateau, with ($L,k,t^*$) mapped from $x$ to capture individual heterogeneity. (Visualizations of the curves: https://freeimage.host/i/B2EWRN1)
> |Method\ (STS)|N=150($\mathcal{R}$)($10^{-2}$)|N=250($\mathcal{R}$)($10^{-2}$)|N=350($\mathcal{R}$)($10^{-2}$)|N=150($\mathcal{E}_{dose}$)($10^{-1}$)|N=250($\mathcal{E}_{dose}$)($10^{-1}$)|N=350($\mathcal{E}_{dose}$)($10^{-1}$)|
> |:---|:---|:---|:---|:---|:---|:---|
> |TS|1.19±1.27|0.89±0.96|0.64±0.83|1.31±0.20|1.29±0.21|1.27±0.17|
> |CATS|1.21±1.03|0.53±0.41|0.28±0.38|**1.22±0.18**|**1.15±0.13**|*1.16±0.21*|
> |ABC3|1.13±1.37|*0.37±0.54*|**0.16±0.40**|1.32±0.27|1.22±0.20|1.18±0.13|
> |SoftTopK|**0.56±0.38**|0.73±0.70|0.66±0.54|1.28±0.23|1.30±0.28|1.20±0.22|
> |GVALID|0.76±0.92|**0.36±0.47**|*0.17±0.36*|1.27±0.22|*1.15±0.19*|**1.09±0.14**|
> - Full results: https://freeimage.host/i/B2EWuHB
>
> Results show GVALID achieves regret comparable to baselines, demonstrating acceptable generalization when the assumption is violated. However, in clinical practice, while multiple doses on a plateau can yield optimal regret, identifying the minimum effective dose to minimize toxicity is important [1]. Thus, the optimal dose is practically defined as the plateau's inflection point. Evaluated on this optimal dose error $\mathcal{E}_{dose}$, GVALID outperforms existing methods. This precision likely stems from our gradient-based surrogate's sensitivity to local curvature, pinpointing the optimum more effectively than purely exploratory approaches. We hope these results evidence our method's broader applicability.
>
> **Finite-difference construction:**
> We clarify that the finite-difference $\tilde{g}$ is computed using the GP posterior mean $\mu(x,t)$ to estimate potential outcomes. This is a purely computational step on the surrogate model, requiring no multiple physical interventions on the same subject.
>
> **Minor comments:**
> We have standardized all figure color and emphasized the definition of $f$.
>
> **Other type of outcome:**
> Thank you for this important point. GVALID currently targets continuous outcomes, as its gradient-based objective requires a smooth, differentiable dose-response function. We will state this limitation in the revision. Extending it to binary endpoints, potentially by applying our objective to a GP classifier's latent space via a link function, remains a promising direction for our future work.
>
> **Global estimation:**
> To evaluate GVALID's global estimation under potential model mismatch, we introduce the metric $\mathcal{E}_ {\text{curve}}:=\mathbb{E}_ x\left[\int_ 0^1(\mu(x,t)-f(x,t))^2dt\right]$, computed via test-set averaging and grid approximation. Results on both unimodal (News) and plateau-shaped (STS) datasets are shown below.
> |Method\ $\mathcal{E}_{curve}$|N=150(News)|N=250|N=350|N=150(STS)|N=250|N=350|
> |:---|:---|:---|:---|:---|:---|:---|
> |TS|3.24±3.33|2.81±3.10|2.15±2.69|0.31±0.09|0.29±0.08|0.29±0.09|
> |CATS|1.06±0.68|0.85±0.47|0.79±0.56|0.28±0.14|0.26±0.09|0.25±0.09|
> |ABC3|*0.59±0.26*|*0.45±0.15*|0.39±0.15|0.17±0.04|0.16±0.02|0.15±0.01|
> |SoftTopK|4.19±4.23|3.87±4.38|3.49±4.39|**0.15±0.02**|**0.14±0.01**|**0.13±0.01**|
> |GVALID|**0.54±0.21**|**0.42±0.14**|**0.34±0.10**|0.26±0.08|0.25±0.08|0.23±0.07|
>
> Although GVALID slightly trails estimation-focused baselines in $\mathcal{E}_{curve}$ on STS, this aligns with our goal: prioritizing the sampling budget for swift optimal dose identification over global fitting. By strategically sacrificing accuracy in non-optimal regions, it maintains high precision near the optimum (Example: https://freeimage.host/i/B2EWzcQ (News) & https://freeimage.host/i/B2EWciJ (STS)). Ultimately, GVALID remains effective at its primary policy optimization task across all scenarios.
>
> **Dose-response literature:**
> As requested, we included [2] to justify our smoothness assumption and Bayesian modeling, the more recent [3] to further support Assumption 4.1.
>
> [1]. Guo, B., et al. "DROID: dose-ranging approach to optimizing dose in oncology drug development."
>
> [2]. Hamza, T., et al. "A Bayesian dose-response meta-analysis model: A simulations study and application."
>
> [3]. Taleb, N., et al. "Working with convex responses: Antifragility from finance to oncology." Entropy (2023).

---

> > ### Author Rebuttal · Reviewer_yJDK · 2026-04-02
> >
> > Thank you for your helpful replies. My concerns have been appropriately addressed. The added theory and ablations exploring plateauing dose-response relationships and GVALID’s global estimation strengthen the paper. I am also pleased with the authors’ plan to contextualize their work with further literature from the trial community. I have decided to increase my score to 4 (weak accept).

---

> > > ### Author Response · Authors · 2026-04-02
> > >
> > > Thank you for your constructive review and for carefully reading our rebuttal. We are glad that the added theory and empirical validations on plateauing dose-response relationships and GVALID's global estimation have addressed your concerns. We will also incorporate additional literature from the trial community as discussed.
> > >
> > > We also wanted to kindly note that the updated score may not yet be reflected in the system. This could be due to a delay, but we wanted to flag it just in case. We truly appreciate your time and effort throughout the review process!

---

### Official Review · Reviewer_CXdr · 2026-03-10

**Soundness:** 3
**Presentation:** 3
**Significance:** 3
**Originality:** 3
**Overall Recommendation:** 4
**Confidence:** 3

**Summary:**

The paper studies active policy optimization for learning individualized dosing strategies (IDSs). The idea is to combine active learning with policy gradient algorithms for estimating IDSs. In particular, it proposes to select candidate samples by minimizing the variance of the resulting policy gradient. Theoretically, the author(s) established a result that links the regret of the final policy with the variance of the policy gradient to motivate their proposal. Empirically, a comprehensive numerical study is conducted, involving (i) comparison against a large number of baseline approaches; (ii) synthetic data and semi-synthetic real-world datasets; (iii) ablation studies; (iv) sensitivity analyses and (v) empirical validations of the theoretical results.

**Compliance With Llm Reviewing Policy:**

Affirmed.

**Final Justification:**

As mentioned in my review, my concerns have been addressed. I will maintain my acceptance score.

**Key Questions For Authors:**

I was wondering whether the author(s) could derive a more rigorous regret bound that explicitly accounts for the multi-step nature of the active learning algorithm. The regret bound in Theorem 4.1 relates regret to the variance of the policy gradient, but the analysis appears to focus only on a single round, assuming that the policy from the previous round is used to estimate the gradient. As a result, the theory does not fully characterize the regret of the final policy obtained after multiple rounds of active learning.

More importantly, the current analysis does not incorporate several practical components of Algorithm 1, such as the batch size, target size, candidate size, or the finite-sample discrepancies that arise during implementation. This creates a gap between the theoretical development, which is relatively simple, and the actual algorithm studied empirically.

**Limitations:**

I did not find relevant discussions of the limitations of the proposed methods.

**Strengths And Weaknesses:**

Strengths:

1. The topic addressed in this paper is relatively underexplored. While there exists a large body of literature on individualized treatment regimes and on active learning in causal inference, the combination of these two directions appears to be less studied. In this regard, the proposed methodology for active learning of optimal dosing rules represents a novel contribution.

2. The numerical experiments are comprehensive. The authors compare their method against a large number of baseline approaches and evaluate performance using both synthetic data and semi-synthetic datasets derived from real-world data. The empirical results also  validate the theoretical findings, and additional ablation and sensitivity analyses are provided as well.

Weaknesses:

1. There is a large literature on individualized dosing rules (e.g., https://www.tandfonline.com/doi/full/10.1080/01621459.2016.1148611, https://www.jmlr.org/papers/volume24/21-0843/21-0843.pdf). Yet, only few work is discussed in the related work section, and the discussion itself remains quite limited.

2. While Assumptions B.1–B.3 are commonly imposed in the literature, Assumption B.4 does not appear to be standard. Given its importance, this assumption should be stated and discussed in the main paper rather than only in the appendix. A clearer explanation of its role and practical feasibility is needed to improve the transparency of the methodological development.

3. Another potentially restrictive assumption is Assumption 4.1. It goes beyond a standard smoothness condition, as suggested by its title. In particular, it requires the negative Hessian to be bounded both above and below. While imposing an upper bound (L) is common in smoothness assumptions, requiring the lower bound ($\mu$) to be bounded away from zero is considerably stronger. This condition implies that, for each realization of the covariate (x), there exists a unique treatment (t) that optimizes the dose. Such a uniqueness assumption may be unrealistic: In many medical applications, there may exist subsets of patients whose outcomes are largely insensitive to a range of treatment options, leading to multiple near-optimal treatments (see, e.g., https://arxiv.org/abs/1603.07573).

4. Some notations appear without clear definition. For example, the function g in Eqn. (9) is not explicitly defined. While it seems to correspond to the estimated gradient, this should be clearly stated.

---

> ### Author Rebuttal · Authors · 2026-03-30
>
> Thank you for your review.
>
> **Literature on individualized dosing:**
>
> We appreciate the suggested references. However, unlike existing literature focused on offline learning from static data, our work addresses active interventional data acquisition. We will incorporate these citations and clarify this distinction in the revised manuscript.
>
> **Assumption B.4:**
>
> Assumption B.4 ensures $\mathbb{E}[g(x, \hat t_D^\star(x)) \mid D] = 0$, simplifying the second moment to a pure posterior variance. As supported by [1], optimal doses are typically interior points to avoid boundary toxicity. We will provide a detailed explanation of this in the revised main text.
>
>
> **Assumption 4.1:**
>
> We agree with the reviewer's insight.
>
> While our initial analysis assumed a globally unique optimum, GVALID's core mechanism generalizes to broader settings. To accommodate realistic scenarios, we extend our theory by: (1) relaxing global concavity to a Local Error Bound; and (2) allowing the optimal dose to be a flat region.
>
> *Relaxed Assumption 4.1' (Local Error Bound):*
> *Let the set of optimal treatments for a given covariate $x$ be $\mathcal{T}^\star(x):=\arg\max_{t \in \mathcal{T}} f(x,t)$. We assume that for each $x$, there exists a local neighborhood $U_x$ around $\mathcal{T}^\star(x)$ such that for any $t \in U_x$:*
>
> *1). $R(x;t) \le \frac{L}{2} \operatorname{dist}(t, \mathcal{T}^\star(x))^2$*
>
> *2). $\operatorname{dist}(t, \mathcal{T}^\star(x)) \le \frac{1}{\mu} |g(x,t)|$*
>
> *(Note: $\operatorname{dist}$ denotes the shortest distance to the optimal set.)*
>
> Under this relaxation, the core relationship between regret and gradient variance still holds, with the addition of a penalty term for early-stage exploration.
>
> *Relaxed Theorem 4.1': Under Assumption 4.1', the Bayes regret of the posterior mean policy is bounded by:*
> $$R(\mathcal{D}) \le \frac{L}{2\mu^2} \mathbb{E}_ X\big[ \operatorname{Var}( g(X, \hat t_{\mathcal{D}}^* (X)) \mid \mathcal{D} ) \big] + \xi_ {\mathcal{D}},$$
> *where $\xi_{\mathcal{D}}$ is the escape penalty*
> $$\xi_{\mathcal{D}} := \mathbb{E}_ X\Big[ R\big(X; \hat t_{\mathcal{D}}^* (X)\big) \mathbf{1}[\hat t_{\mathcal{D}}^* (X) \notin U_X ] \mid \mathcal{D} \Big].$$
>
> Given the bounded domain $\mathcal{T}$ and smooth $f(x,t)$, the escape penalty $\xi_{\mathcal{D}}$ is bounded and primarily reflects early-stage exploration costs. This theoretically justifies our use of a warm start: collecting a slightly larger initial dataset to fit the GP minimizes $\xi_{\mathcal{D}}$ early on by keeping estimates within the effective neighborhood $U_X$. Consistent with our SWV results, this strategy helps the algorithm quickly escape suboptimal regions, allowing gradient-variance reduction to drive the optimization.
>
> We will incorporate this relaxed analysis into our manuscript as formal proposition and remark.
>
> Furthermore, for new experimental results on plateau-shaped curves, please see our response to Reviewer yJDK.
>
> **Cumulative regret and dependencies:**
>
> *Theorem (Average Cumulative Regret.) Under Assumptions 4.1, 4.2, for any $\eta \in (0,1)$, the average cumulative simple regret of GVALID over $T$ rounds satisfies, with probability at least $1-\eta$:$$ \frac{1}{T} \sum_ {\tau=0}^{T-1} R(D_\tau) \le C_g \Bigg[ \frac{1}{\rho_B m T} \tilde\Phi_ {D_0}(Z_0) + \frac{1}{\rho_B m T} \sum_ {\tau=0}^{T-1} \big( E^{\mathrm{cand}}_ \tau + \Gamma_\tau + \Lambda_\tau \big) + \beta_{\mathrm{fd}}(\delta) + \beta_{\mathrm{grid}}(h_c) + G_m^2 \sqrt{\frac{\log(2T/\eta)}{2m}} \Bigg] $$*
>
> Here, $\rho_B \in (0,1)$ is the variance contraction rate, which increases with $B$ due to submodularity. The terms $E^{\mathrm{cand}}_ \tau := \sum_ {b=1}^{B_\tau} \big( \max_{q \in \overline{\mathcal{Q}}} \Delta_{\tau,b}(q) - \max_{q \in \mathcal{Q}} \Delta_{\tau,b}(q) \big)$, $\Gamma_ \tau := \tilde\Phi_{D_{\tau+1}}(Z_{\tau+1}) - \tilde\Phi_{D_{\tau+1}}(Z_\tau)$, and $\Lambda_ \tau := \tilde\Phi_ {D_{\tau+1}}(Z_\tau) - \tilde\Phi^{S_\tau}_ {D_\tau}(Z_\tau)$ denote the subsampling error, target drift, and model-update discrepancy, respectively—all proven to decay during active learning. Finally, $\beta_{\mathrm{fd}}(\delta) = \mathcal{O}(\delta^2)$ and $\beta_{\mathrm{grid}}(h_c) = \mathcal{O}(h_c)$ represent the finite-difference and grid discretization errors.
>
> *Proof Sketch: By leveraging the submodularity of greedy batch search to ensure a constant variance contraction rate $\rho_B$, we sum the discounted structural perturbations over $T$ rounds to bound the cumulative regret up to irreducible errors.*
>
> For a fixed budget $N=TB$, the term $B/N$ highlights a core trade-off: a larger $B$ improves the contraction rate $\rho_B$, but fewer updates $T$ amplify structural perturbations.
>
>
> **Regarding notation:**
>
> We have ensured consistent symbol definitions throughout the revised manuscript.
>
> [1]. Yuan, Ying, et al. "Statistical and practical considerations in planning and conduct of dose-optimization trials." Clinical Trials 21.3 (2024): 273-286.

---

> > ### Author Rebuttal · Reviewer_CXdr · 2026-04-03
> >
> > I appreciate the author(s)' efforts made during the rebuttal. My concerns have been addressed. I will maintain my acceptance score and will contribute to the AC–reviewer discussion as appropriate.

---

> > > ### Author Response · Authors · 2026-04-04
> > >
> > > Thank you for your careful evaluation and for acknowledging our efforts during the rebuttal. We sincerely appreciate your constructive feedback and continued support.

---

### Official Review · Reviewer_PnEa · 2026-03-13

**Soundness:** 3
**Presentation:** 3
**Significance:** 2
**Originality:** 3
**Overall Recommendation:** 4
**Confidence:** 3

**Summary:**

This paper proposes GVALID, a batch active learning method for individualized dosing in a one-shot cohort setting. The main idea is to select samples that are most useful for improving the dosing policy itself, rather than for globally fitting the response surface. The paper derives a regret bound in terms of posterior gradient uncertainty at the estimated optimal dose, and then uses a greedy batch construction based on a finite-difference approximation of that quantity. The empirical study includes three synthetic datasets and one semi-synthetic dataset, with comparisons to a fairly broad set of baselines.

**Compliance With Llm Reviewing Policy:**

Affirmed.

**Key Questions For Authors:**

Can you provide a sensitivity analysis for the finite-difference step size delta, or explain more clearly why exact GP derivative posteriors were not used where available?

Since SWV violates the main assumptions, can you provide a more systematic analysis of how performance changes as those assumptions break down?

What would be required to evaluate the method in a more realistic dosing setting, even retrospectively, rather than only in synthetic or semi-synthetic environments?

Can you add formal paired statistical tests or effect sizes for the main empirical comparisons?

**Limitations:**

The paper does acknowledge that SWV is outside the clean theoretical regime, but the main paper should more clearly emphasize that the evaluation remains synthetic or semi-synthetic and does not yet validate the method in a realistic clinical setting.

**Strengths And Weaknesses:**

I think the paper has a real methodological contribution. The main strength is that the acquisition rule is tied to the actual decision objective, which is more compelling here than generic uncertainty reduction. The theoretical development is also meaningful rather than decorative, and the empirical results are generally strong. The ablations are helpful as well: they make it clear that both the gradient-based criterion and the joint optimization over covariates and dose are doing real work.

My main concern is that the evidence is not yet as strong as the framing. One issue is that the theory is built around fairly regular dose-response structure, while SWV explicitly violates that regime and is treated more as a stress test than as a setting where the limits of the theory are really analyzed. A second issue is that the method relies on a finite-difference approximation to the gradient, but I did not find a convincing sensitivity analysis for that choice. More broadly, despite the clinical motivation, the evaluation is still synthetic or semi-synthetic, so I do not think the paper currently supports a strong claim about practical individualized dosing. Finally, the paper reports standard deviations, but given the number of baselines and budgets, I would have liked to see formal significance testing.

---

> ### Author Rebuttal · Authors · 2026-03-30
>
> Thank you for your thorough review and appreciation of our work. Below, we address your main concerns.
>
> **Regarding more systematic analysis of when assumptions break down:**
>
> When assumptions break down, an initial warm start effectively helps overcome the escape penalty of suboptimal or flat regions, as theoretically justified by our newly introduced local error bound. Please see our response to Reviewer CXdr for the detailed analysis.
>
> **Regarding more realistic dosing setting:**
>
> We fully agree with the reviewer that clinical validation is the ultimate goal. However, due to the ethical risks of active interventions and the lack of observable counterfactuals in real-world data, synthetic benchmarks still remain the standard for regret evaluation [1]. To ensure genuine biomedical relevance despite these constraints, our simulations are grounded in established biological models [2]. Moving forward, we aim to validate our method within interactive, high-fidelity clinical simulators to further enhance its practical significance. Accordingly, we have tempered our claims in the revised manuscript to accurately reflect these current limitations.
>
> [1]. Cha, Taehun, and Donghun Lee. "Abc3: Active bayesian causal inference with cohn criteria in randomized experiments." Proceedings of the AAAI Conference on Artificial Intelligence. Vol. 39. No. 25. 2025.
>
> [2]. Guo, Beibei, and Ying Yuan. "DROID: dose-ranging approach to optimizing dose in oncology drug development." Biometrics 79.4 (2023): 2907-2919.
>
> **Regarding the choice of finite-difference approximation:**
>
> We opted for a finite-difference approximation over exact GP derivatives for two main reasons, which are now supported by a sensitivity analysis in the revision.
>
> 1. Computational Efficiency: Exact GP gradients require costly backpropagation through the computational graph. Our finite-difference approach strictly uses two parallelizable forward passes, running 4-5x faster in practice.
> 2. Robustness via Smoothing: In early active learning stages, GP hyperparameters are often unstable, making exact analytical gradients highly noisy. Finite-differences inherently act as a spatial smoothing filter, improving candidate selection.
> 3. Sensitivity Analysis on $\delta$ : Our results reveal a clear tradeoff: too small a $\delta$ (or exact gradients) amplifies GP estimation noise, whereas too large a $\delta$ introduces excessive truncation error. Moderate values optimally balance noise filtering and gradient fidelity to yield the best performance.
>
> | Method (GVALID) | N=150 (CSC) | N=250 (CSC) | N=350 (CSC) |
> | :--- | :--- | :--- | :--- |
> | $\delta = 0.005$ | 1.38 ± 2.27 | 0.85 ± 1.52 | 0.65 ± 1.24 |
> | $\delta = 0.01$  | 0.91 ± 0.46 | 0.58 ± 0.27 | 0.45 ± 0.22 |
> | $\delta = 0.02$ | **0.77 ± 0.40** | 0.54 ± 0.33 | **0.39 ± 0.20** |
> | $\delta = 0.05$ | 1.15 ± 1.07 | 0.53 ± 0.33 | 0.41 ± 0.23 |
> | $\delta = 0.1$ | 0.86 ± 0.55 | **0.50 ± 0.29** | 0.40 ± 0.24 |
> | **True Grad** | 1.56 ± 0.80 | 1.26 ± 0.53 | 1.09 ± 0.45 |
>
> A complete comparison of the convergence curves can be found at the following link:
> https://freeimage.host/i/B2EWYog
>
> **Regarding significance testing:**
>
> We have incorporated paired Wilcoxon signed-rank tests into the main empirical results in Table 1. The results confirm that our proposed method achieves a statistically significant advantage ($p < 0.01$) over all baselines across all evaluated datasets in the later stages of the process. As an illustrative example, the detailed results for the HNL dataset are presented below.
>
> | METHOD | N=150 (HNL)| N=250 (HNL)| N=350 (HNL)|
> | :--- | :--- | :--- | :--- |
> | TS | 0.56±.21 * | 0.39±.12 ** | 0.31±.11 ** |
> | GP-UCB | 0.78±.38 ** | 0.58±.27 ** | 0.46±.16 ** |
> | EI | 0.90±.35 ** | 0.68±.23 ** | 0.55±.20 ** |
> | CATS | 0.71±.27 ** | 0.43±.11 ** | 0.31±.08 ** |
> | ABC3 | 0.43±.15 | 0.24±.09 ** | 0.17±.05 ** |
> | PG | 0.54±.18 * | 0.40±.09 ** | 0.33±.07 ** |
> | **GVALID** | **0.40±.24** | **0.18±.09** | **0.12±.04** |
>
> - Statistically significant differences compared to our method are highlighted using *, with single and double symbols denoting $p < 0.05$ and $p < 0.01$, respectively. For brevity, only competitive methods are reported.

---

> > ### Author Rebuttal · Reviewer_PnEa · 2026-04-06
> >
> > the relaxed Theorem 4.1' and cumulative regret decomposition in the CXdr thread, plus the Wilcoxon tests and δ sensitivity, address most of my concerns; the lack of a real clinical dataset is the one gap that remains, but I appreciate the tempered claims and will keep my score.

---

> > > ### Author Response · Authors · 2026-04-06
> > >
> > > We are glad the rebuttal updates addressed your concerns. We sincerely appreciate your continued support and your thorough review, which has genuinely improved our work.

---

### Official Review · Reviewer_3vfs · 2026-03-14

**Soundness:** 3
**Presentation:** 3
**Significance:** 3
**Originality:** 3
**Overall Recommendation:** 4
**Confidence:** 4

**Summary:**

This paper proposes GVALID, an active learning method for learning individualized continuous dosing policies under a limited experimental budget. The core idea is to upper-bound the final policy regret by the posterior variance of the dose gradient evaluated near the currently estimated optimum, and then to choose new interventions so as to reduce that uncertainty as efficiently as possible.

**Compliance With Llm Reviewing Policy:**

Affirmed.

**Key Questions For Authors:**

1. Can you derive any lower bound, impossibility result, or instance-dependent complexity characterization for policy learning in your continuous-dose setting? If not, how should the reader judge whether minimizing the proposed upper bound is statistically near-optimal?

2. How does GVALID compare, both conceptually and empirically, with adaptive experimental design and best-arm identification approaches to policy learning, especially Kato et al. (2024) and related pure-exploration bandit methods?

3. What is the precise advantage of gradient-variance minimization over more direct decision-focused criteria, such as mis-selection risk, local gap estimation, or other surrogates for final policy error?

4. Given the fixed-budget terminal-policy objective, why are pure-exploration / contextual BAI baselines not more central in the empirical study?

5. Since the method ultimately relies on discretization and finite-difference approximations in the dose dimension, how much of the empirical gain is due to the gradient-based acquisition rule itself, rather than the GP surrogate and the discretized optimization procedure?

**Limitations:**

The paper is discussed mainly from a causal-inference perspective, but I believe its closest methodological neighborhood is bandits—especially pure exploration, best-arm identification, contextual policy learning, and continuous-action contextual bandits. A substantially deeper review of that literature is necessary; otherwise, it is difficult to tell what is genuinely new here and what is mainly a repackaging of existing decision-focused sampling ideas for the individualized dosing setting (Bubeck et al., 2011, Pure Exploration in Finitely-Armed and Continuous-Armed Bandits, Theoretical Computer Science; Kato et al., 2024, Adaptive Experimental Design for Policy Learning, arXiv/OpenReview; Krishnamurthy et al., 2020, Contextual Bandits with Continuous Actions: Smoothing, Zooming, and Adapting, JMLR).

**Strengths And Weaknesses:**

The paper addresses an important and practically relevant problem. The focus on individualized continuous dosing under budget and one-shot intervention constraints is meaningful, the gradient-based viewpoint is interesting, and the empirical section is reasonably broad.

My main concern is novelty and positioning. At a high level, the paper translates final decision quality into a local uncertainty quantity around the estimated optimum and then designs the sampling rule to reduce that quantity. This decision-focused experimental-design template is already familiar in pure-exploration and best-arm identification work, where sample allocation is driven by the goal of reducing simple regret or final decision error rather than improving global prediction accuracy (Bubeck et al., 2011, Pure Exploration in Finitely-Armed and Continuous-Armed Bandits, Theoretical Computer Science; Kaufmann et al., 2016, On the Complexity of Best Arm Identification in Multi-Armed Bandit Models, JMLR; Fiez et al., 2019, Sequential Experimental Design for Transductive Linear Bandits, NeurIPS; Jedra and Proutiere, 2020, Optimal Best-arm Identification in Linear Bandits, NeurIPS). In that sense, the main high-level principle does not yet strike me as fundamentally new; the novelty appears to lie in the specific gradient-variance surrogate, and the paper should isolate that contribution more clearly.

Since the evaluation target is the quality of the final recommended policy, the closest theoretical lens seems to be simple-regret / pure-exploration bandits rather than standard cumulative-regret bandits (Bubeck et al., 2011, Pure Exploration in Finitely-Armed and Continuous-Armed Bandits, Theoretical Computer Science). However, the experimental comparisons are dominated by exploration-exploitation baselines such as TS/UCB/EI-style methods. I would have liked a stronger comparison to pure-exploration or best-arm-identification-oriented designs.

Relatedly, adaptive experimental design for policy learning has already been formalized as contextual best-arm identification, with lower bounds and asymptotic optimality guarantees, in Kato et al. (2024, Adaptive Experimental Design for Policy Learning, arXiv/OpenReview). Because of that, the present paper currently reads more like a continuous-dose specialization of a known bandit-design principle than a fundamentally new conceptual contribution.

I agree that focusing on gradient uncertainty is an interesting technical choice. However, the paper does not yet explain clearly why gradient variance is preferable to more direct decision-error surrogates from related active design or bandit literatures. More broadly, choosing samples to minimize a downstream error surrogate is itself not new; for example, ALICE explicitly selects samples by targeting the conditional expectation of generalization error rather than global uncertainty alone (Sugiyama, 2006, Active Learning in Approximately Linear Regression Based on Conditional Expectation of Generalization Error, JMLR). The paper would be much stronger if it clarified what is uniquely gained by optimizing gradient variance specifically.

A second major concern is the lack of a matching lower bound or an instance-dependent difficulty characterization under the paper's assumptions. In closely related pure-exploration and best-arm identification literatures, lower bounds are crucial because they tell us whether an acquisition rule is near-optimal or merely heuristic (Kaufmann et al., 2016, On the Complexity of Best Arm Identification in Multi-Armed Bandit Models, JMLR; Jedra and Proutiere, 2020, Optimal Best-arm Identification in Linear Bandits, NeurIPS; Kato et al., 2024, Adaptive Experimental Design for Policy Learning, arXiv/OpenReview). Here, the theory mainly justifies the method through upper-bound minimization, so it is difficult to assess how informative or tight that surrogate actually is.

Finally, although the empirical section includes several bandit baselines, the conceptual positioning is still too centered on active learning for causal inference. Methodologically, this work seems at least as close—arguably closer—to continuous-action contextual bandits, pure exploration, and adaptive experimental design than to the mainstream causal-effect-estimation literature (Krishnamurthy et al., 2020, Contextual Bandits with Continuous Actions: Smoothing, Zooming, and Adapting, JMLR; Majzoubi et al., 2020, Efficient Contextual Bandits with Continuous Actions, NeurIPS; Zhu and Mineiro, 2022, Contextual Bandits with Smooth Regret, ICML). Without a deeper comparison to that literature, it is hard to evaluate the paper's true marginal contribution.

---

> ### Author Rebuttal · Authors · 2026-03-30
>
> **Regarding novelty and positioning:**
>
> We thank you for the assessment of our work’s positioning. We agree that our framework fundamentally aligns with pure-exploration and best-arm identification (BAI) paradigms, serving as a continuous-dose specialization. However, our research is deeply motivated by real-world dose-response policy optimization, a prevalent scenario in causal inference where specific structural properties are widely recognized by domain experts, yet direct continuous policy optimization remains underexplored.
>
> Consequently, rather than claiming a fundamentally new conceptual framework, our core technical novelty lies in the proposed gradient-variance surrogate objective, which is tailored specifically to exploit these continuous structural priors.
>
> In the revision, we tone down our broader motivational claims, use your recommended references to better position our work at the intersection of BAI and causal inference, and explicitly highlight the gradient-variance surrogate as our primary algorithmic contribution.
>
> **Empirical evaluation against BAI methods:**
>
> We initially relied on exploration-exploitation baselines (e.g., TS, UCB, EI) because they are readily available for continuous domains, whereas existing standard pure-exploration algorithms require complex continuous adaptations.
>
> However, we agree that a BAI baseline offers better theoretical alignment. Accordingly, we have incorporated PLAS (Kato et al., 2024), a state-of-the-art contextual BAI method, following your suggestion. To ensure a fair comparison, we adapted the originally discrete PLAS for continuous search by equipping it with the same GP estimator and discretization used in our framework. As shown below, our method consistently outperforms this enhanced PLAS in both adaptive sampling and final decision quality at $N=350$.
>
>
> |Method|N=150(HNL)|N=250(HNL)|N=350(HNL)|
> |:---|:---|:---|:---|
> |PLAS|$0.49\pm0.17^*$|$0.41\pm0.12^{**}$|$0.33\pm0.09^{**}$|
> |GVALID|**0.40$\pm$0.24**|**0.18$\pm$0.09**|**0.12$\pm$0.04**|
>
> **The advantage of our surrogate over direct decision-error surrogates:**
>
> We respectfully suggest that our gradient-variance objective offers certain advantages over other direct decision-focused criteria in both theoretical and empirical aspects.
>
> Theoretically, evaluating direct criteria in continuous domains requires computationally prohibitive multiple integrations. Conversely, our objective provides an efficient closed-form solution and leverages the geometric prior of smooth curves, offering a more direct search mechanism.
>
> Empirically, our ablation study (Table 2) compares GVALID with GVALID-F ( a representative "surrogate for final policy error" minimizing predictive variance at the estimated $t^* $). To ensure a fair comparison, the downstream error estimation for these methods was conducted on the same representative sampled set. GVALID's superior performance reveals a potential problem in other direct criteria: they over-rely on inaccurate early estimates of $t^*$, misguiding exploration. By smoothly tracking gradient uncertainty, our method yielding greater robustness and sample efficiency.
>
> **The lower bound:**
>
> We have derived a minimax regret lower bound for our setting which corresponds to Theorem 4.1 (upper bound). It shows that our framework, by actively minimizing posterior gradient variance, precisely targets the fundamental lower bound of the continuous policy learning problem. This establishes that gradient variance minimization is not a mere heuristic, but a theoretically optimal strategy that directly addresses the intrinsic difficulty $\Lambda_T(X)$.
>
> *Theorem (Minimax Regret Lower Bound.)*
> *Let $\Pi$ be a policy class with Natarajan dimension $M$, shattering a set of contexts $\mathcal{S}\subset\mathcal{X}$ drawn from a marginal distribution $P_X$. Under Assumption 4.1, for any total budget $T$ and any sampling strategy, the worst-case expected policy regret is bounded below by:*
> *$$\sup_ {P\in\mathcal{P}}\mathbb{E}_ P[R(\hat{\pi})]\ge C\cdot \mathbb{E}_ {X\sim P_ X}\left[\frac{\mu}{L^2}\frac{M\sigma_ \epsilon^2}{T\cdot\Lambda_ T(X)}\right], $$*
> *where $C>0$ is an constant, and the normalized allocation spread $\Lambda_T(X)$ is defined as*
> $\Lambda_T(X):=\frac{M}{T}V_T(X)=\frac{M}{T}\mathbb{E}_ P \left[\sum_ {i =1}^T(t_ i-t^* (X))^2\mathbb{I}[x_i=X]\right].$
>
> *Remark. When estimating the target gradient $g(X,t)$ in the local neighborhood of the optimal dose $t^{\star}$ via GP, the posterior gradient variance is proportional to the allocation spread $V_T(X)$:*
> $$\text{Var}(g(X,t^{\star} (X))|\mathcal{D})\propto \frac{\sigma_ \epsilon^2}{V_T(X)}.$$
>
>
> **The source of empirical gains:**
>
> To ensure a fair comparison, all baselines were standardized using the identical GP surrogate and discretization grid (see Appendix G). Consequently, the observed improvements are directly attributable to our gradient-based acquisition rule, rather than the underlying estimator or optimization procedure.

---

> > ### Author Rebuttal · Reviewer_3vfs · 2026-04-07
> >
> > Thank you for your reply. I have a question: while the authors indicate in their reply that the order of regret is T, shouldn’t the order of minimax regret be $\sqrt{T}$? For example, the Natarayan dimension-dependent minimax lower bounds derived in Theorem 3.3--3.4 of Kato, Okumura, Ishihara, and Kitagawa (2024) “Adaptive Experimental Design for Policy Learning” (though I’m not sure if this is correct since the paper appears to be a preprint), and the lower bound for binary treatments in Kitagawa and Tetenov (2018), it is indeed $\sqrt{T}$. I would appreciate it if you could clarify why it is $T$.

---

> > > ### Author Response · Authors · 2026-04-08
> > >
> > > We appreciate this point. As established in prior work, the minimax regret lower bound for general policy learning is indeed $\Omega(1/\sqrt{T})$.
> > >
> > > However, the difference in our regret order results from our specific problem setting:
> > >
> > > **The $\Omega(1/T)$ Lower Bound:**
> > > Our $\Omega(1/T)$ simple regret lower bound relies on the strong concavity assumption. This structure significantly reduces complexity compared to general policy learning, enabling a faster rate. Similar $\mathcal{O}(1/T)$ bounds are established in the literature for problems under similar settings (e.g., Theorems 1 and 3 in [1]). Note that this bound characterizes the inherent problem difficulty assuming access to true gradients.
> > >
> > > **Upper Bound and Estimation Errors:**
> > > Since our algorithm relies on gradient estimation, we do not claim to strictly achieve this tight $\mathcal{O}(1/T)$ bound. As shown in our *Theorem (Average Cumulative Regret)* (see our response to Reviewer CXdr), our upper bound retains bias terms ($\beta_{fd}$ and $\Gamma_{\tau}$) from gradient estimation errors. We have proven their sub-linear convergence, but determining their exact order is left for future work.
> > >
> > > **Corroboration from Related Work:**
> > > Additionally, [2] studies a similar problem using a gradient-free mirror descent method. By approximating gradients via randomized function differences, they derive an $\tilde{\mathcal{O}}(1/T)$ simple regret bound under strong convexity (see Table 2). This further corroborates the plausibility of faster rates under these specific structural conditions.
> > >
> > > We will provide further clarification regarding these specific bounds and settings in the revised version.
> > >
> > > [1].Hazan, Elad, and Satyen Kale. "Beyond the regret minimization barrier: optimal algorithms for stochastic strongly-convex optimization." The Journal of Machine Learning Research 15.1 (2014): 2489-2512.
> > >
> > > [2].Gasnikov, Alexander V., et al. "Stochastic online optimization. Single-point and multi-point non-linear multi-armed bandits. Convex and strongly-convex case." Automation and remote control 78.2 (2017): 224-234.

---

### Decision · Program_Chairs · 2026-04-30

**Decision:**

Accept (regular)

**Comment:**

I concur with the reviewers' uniformly positive view of the submission and recommend weak accept. The paper studies an interesting and underexplored problem, and the reviewers found the decision-focused acquisition rule, the theory linking regret to gradient uncertainty, and the empirical study to be meaningful strengths. The main points to address in the final version are clearer positioning relative to pure exploration, adaptive experimental design, and individualized dosing literatures, more explicit discussion of the restrictive dose-response assumptions and practical scope, and clarification of the gap between the theory and the full multi-round algorithm. The rebuttal addressed many concerns and I trust the relevant clarifications will be incorporated in the final version.